# Cellular arrangement impacts metabolic activity and antibiotic tolerance in *Pseudomonas aeruginosa* biofilms

**Hannah Dayton[1], Julie Kiss[1], Mian Wei[2], Shradha Chauhan[1], Emily LaMarre[3], William Cole Cornell[1], Chase J. Morgan[1], Anuradha Janakiraman[3], Wei Min[2], Raju Tomer[1], Alexa Price-Whelan[1], Jasmine A. Nirody[4], Lars E. P. Dietrich** [1] *

**1** Department of Biological Sciences, Columbia University, New York, New York, United States of America,
**2** Department of Chemistry, Columbia University, New York, New York, United States of America, **3** Program in Biology, The Graduate Center, City University of New York, New York, New York, United States of America,
**4** Department of Organismal Biology and Anatomy, University of Chicago, Chicago, Illinois, United States of America

\* LDietrich@columbia.edu

## Abstract

Cells must access resources to survive, and the anatomy of multicellular structures influences this access. In diverse multicellular eukaryotes, resources are provided by internal conduits that allow substances to travel more readily through tissue than they would via diffusion. Microbes growing in multicellular structures, called biofilms, are also affected by differential access to resources and we hypothesized that this is influenced by the physical arrangement of the cells. In this study, we examined the microanatomy of biofilms formed by the pathogenic bacterium *Pseudomonas aeruginosa* and discovered that clonal cells form striations that are packed lengthwise across most of a mature biofilm's depth. We identified mutants, including those defective in pilus function and in O-antigen attachment, that show alterations to this lengthwise packing phenotype. Consistent with the notion that cellular arrangement affects access to resources within the biofilm, we found that while the wild type shows even distribution of tested substrates across depth, the mutants show accumulation of substrates at the biofilm boundaries. Furthermore, we found that altered cellular arrangement within biofilms affects the localization of metabolic activity, the survival of resident cells, and the susceptibility of subpopulations to antibiotic treatment. Our observations provide insight into cellular features that determine biofilm microanatomy, with consequences for physiological differentiation and drug sensitivity.

**Data Availability Statement:** All relevant data are within the paper and its Supporting Information

## Introduction

Chemical gradients inevitably form during multicellular growth, leading to the establishment of distinct subzones—with different resource availabilities—in cellular aggregates and complex organisms [1]. The conditions of these subzones are important because they promote physiological differentiation, in turn affecting morphogenesis, metabolic activity, and susceptibility to drugs [2–6]. Capacities for substrate uptake and transport are therefore critical features of a

files. All code is available at https://github.com/jnirody/biofilms.

**Funding:** This work was supported by a National Institute of Health (NIH) Diversity supplement to J. K. and by the parent NIH/NIAID grant, R01AI103369, to L.E.P.D. It was also supported by NIH (R01EB029523) and Chan Zuckerberg Initiative (Dynamic Imaging 2023-321166) grants to W.M., a Rockefeller University Physics and Biology Fellowship and an All Souls College Fellowship in Life Sciences to J.A.N., National Science Foundation funding (MCB 2216676 ) to A. J., and NIH funding (DP2MH119423 and R44MH116827) to R.T. The funders had no role in study design, data collection and analysis, decision to publish, or preparation of the manuscript.

**Competing interests:** The authors have declared that no competing interests exist.

**Abbreviations:** LB, lysogeny broth; LPS, lipopolysaccharide; PI, propidium iodide; SEM, scanning electron microscopy; SRS, stimulated Raman scattering.

cellular aggregate that are linked to structural development and our ability to treat disease, with diverse mechanisms controlling these processes in various multicellular structures and organisms. (We use the term "resource" to refer to any nutrient or electron acceptor that contributes to cellular metabolic activity and the term "substrate" to refer to any molecule or particle that can enter and be distributed within a multicellular structure.) A conceptual example that highlights the importance of substrate uptake and transport across multicellular structures is that of the circulatory system, which forms conduits that deliver water and other resources to cells in many macroscopic eukaryotes [7,8]. Circulatory systems allow for efficient nutrient allocation, enabling multicellular organisms to grow, develop, and maintain homeostasis.

The predominant multicellular structure of the microbial world is the biofilm, an aggregate of cells attached to each other and encased in a self-produced matrix [9]. Biofilms often form at interfaces and are subject to steep external and internal chemical gradients [10–14], raising questions about how resources are allocated efficiently within these structures [15]. Studies in diverse microbes have shown that, under controlled conditions, their biofilms follow reproducible stages of morphogenesis, establishing characteristic physical patterning at macroscopic and/or microscopic scales [16–26]. Therefore, although some evidence suggests that passive diffusion through the matrix is a significant mechanism of resource distribution within biofilms [27,28], biofilm populations also have the potential to influence uptake and transport by modulating biofilm structure formation and microanatomy [29–31].

Effects on uptake and transport determine nutrient and electron acceptor availability and thereby directly influence metabolic activity, physiological status, and survival within biofilms. However, the ability of both large and small substrates to enter and travel through biofilms also has consequences for biofilm ecology, macrostructure, and susceptibility to antimicrobials. Biofilm ecology and macrostructure are affected when the abilities of relatively large particles such as other microbes and sediment grains to enter a biofilm are determined by its anatomy [32–37]. Antimicrobial susceptibility, on the other hand, can be affected by biofilm anatomy in 2 ways: (i) via its direct influence on the abilities of antimicrobial compounds to enter and become distributed in the biofilm [38,39]; and (ii) via its effects on access to resources and thereby physiological status, which determines susceptibility to antimicrobials because many of these compounds either require investment of cellular energy in order to cross the membrane, or act on processes such as protein or cell wall synthesis [40–42].

In this study, we examined the cellular-level anatomy of macrocolonies formed by the bacterium *Pseudomonas aeruginosa*, which is a major cause of biofilm-based and chronic infections in immunocompromised individuals [43–45]. We found that *P. aeruginosa* biofilms contain vertically arranged zones that differ with respect to cellular orientation and the proximity of related cells. Genetic analyses and microscopic imaging allowed us to identify cell surface components required for this patterning. Importantly, these approaches also revealed that cellular orientation impacts substrate uptake and distribution across the biofilm. Additional analysis via stimulated Raman scattering (SRS) microscopy and an assay for cell death uncovered correlations between cellular arrangement subzones, physiological status, and susceptibility to antibiotic treatment. Collectively, these data link microanatomy to metabolic differentiation—and its consequences for survival and drug resilience—within maturing biofilms.

## Results and discussion

### Mature *P. aeruginosa* biofilms contain vertical, clonal striations

Growth at surface-air interfaces has been used for decades to guide treatment decisions made in the clinical setting and is a focus of research into *P. aeruginosa* biofilm development [46–55]. In particular, our group and others have used macrocolony biofilms as models to study

the molecular links between resource availability and multicellular physiology [12,56–58]; we have discovered a variety of pathways and mechanisms that support the biofilm lifestyle, some of which contribute to pathogenicity and antibiotic tolerance [59,60]. Macrocolony biofilms are formed when 1 to 10 μl of a cell suspension are pipetted onto agar-solidified media and incubated for several days (**Fig 1A**). The macrocolony biofilm model provides a high-throughput, macroscopic readout for the effects of a broad variety of factors—such as polysaccharide production, pilus function, and second messenger levels—that affect *P. aeruginosa* biofilm development [61–63]. It also provides a reproducible system for growing a physiologically differentiated multicellular structure that is amenable to techniques for mapping variations in extra- and intracellular chemistry, gene expression, and metabolic activity in situ [60,64–67].

Despite the significance of the macrocolony model to our understanding of *P. aeruginosa* biology, we lack detailed information about how cells are arranged within these structures and

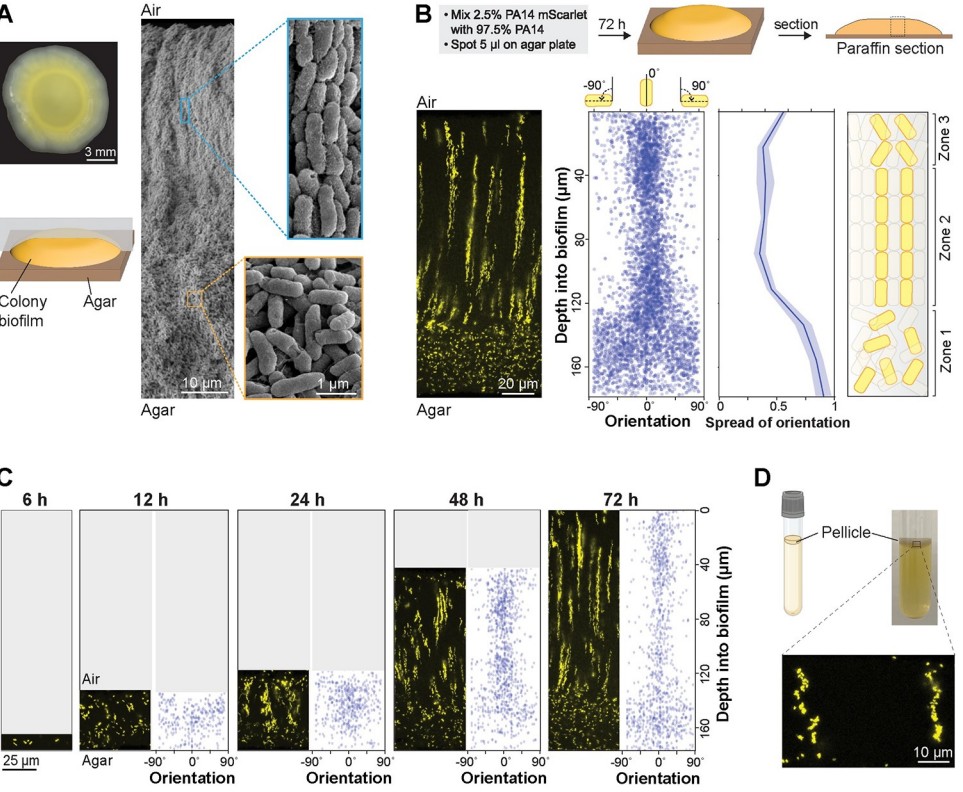

**Fig 1. *P. aeruginosa* cells form vertical striations across depth in colony and pellicle biofilms. (A) Left**: Top view of a *P. aeruginosa* colony biofilm grown for 3 days on 1% tryptone + 1% agar, and schematic showing orientation of the sample used for SEM imaging. **Right**: SEM images of a full colony biofilm cross-section. Insets of higher magnification show cellular arrangement and morphology for the indicated locations in the biofilm. (**B) Top**: Schematic of mixing assay method. **Left**: Fluorescence micrograph of a thin section prepared from a colony biofilm grown in the mixing assay. **Center**: Orientation across depth for fluorescent cells detected in biofilm thin section micrographs. The "spread of orientation" is the standard deviation of orientation values for each pixel across biofilm depth; the values shown in the plot are the average "spread of orientation" at each depth for thin section images taken from 6 biological-replicate biofilms. Shading represents the standard deviation for this average. **Right**: Schematic of cellular arrangement across depth in mature biofilms. (**C**) Micrographs of biofilms prepared as described in (B), but sacrificed at the indicated time points. Scale bar applies to all images. The data underlying Fig 1B and 1C can be found in S1 Data. (**D) Top**: Setup used to grow pellicle biofilms for microscopy. **Bottom**: Fluorescence micrograph of a thin section prepared from a pellicle biofilm. The inoculum contained 2.5% cells that constitutively express mScarlet. Images shown in this figure are representative of at least 2 independent experiments. mScarlet fluorescence is colored yellow. Quantification of colony-forming units confirmed that expression of mScarlet did not affect fitness during growth in mixing assay biofilms (**S1 Fig**).

the genetic determinants of this arrangement. To begin to address this gap, we used scanning electron microscopy (SEM) to examine a WT *P. aeruginosa* PA14 macrocolony biofilm that had grown for 3 days on 1% tryptone, 1% agar. SEM images revealed 2 distinct zones, containing either "disordered" cells (in various orientations) or "ordered" cells (aligned lengthwise) (**Fig 1A**). Similar arrangements of cells aligned lengthwise with ordered packing have been described for diverse bacterial systems [68,69]. Examples include "cords," produced by various mycobacterial species, which contain cells packed along their long axes; and the production of cellulose fibers, by the bacterium *Acetobacter xylinum*, which align with entrapped cells [70,71]. Self-organization has also been reported for *Escherichia coli* cells growing in microfluidic chambers and described for *Vibrio cholerae* colonies [72,73].

To assess the distribution of clonal (related) cells in the ordered region, we grew macrocolony biofilms in a "mixing assay" in which the inoculum contained 2.5% of a strain that constitutively expresses mScarlet as a fluorescent marker ($P_{A1/04/03}$-*mScarlet*). (The remaining 97.5% of the cell population was unmarked.) This mScarlet expression had no effect on fitness in observed biofilms (**S1 Fig**). Biofilms were embedded in paraffin and thin-sectioned, and sections taken from the colony center (**Fig 1A**) were imaged by fluorescence microscopy [74] (**Fig 1B**). We discovered that fluorescent cells within the disordered region (zone 1) are randomly dispersed, while fluorescent cells in the ordered region (zone 2) are arranged in evenly spaced vertical striations reminiscent of the cellular packing seen in the SEM images (**Fig 1A**). Our group has previously observed similar patterning in biofilms formed by a strain in which mScarlet expression is controlled by a stochastically active promoter whose activity status appears to be inherited by daughter cells [75]. To determine the orientations of cells in our mixing-assay biofilms, we used a custom Python script that allowed us to detect individual cells and plot their angles relative to the vertical axis across biofilm depth. While the fluorescent cells in zone 1 are randomly oriented, those in zone 2 are oriented along the vertical axis, consistent with the arrangements visible by SEM (**Fig 1A**). Finally, in the uppermost portion of the biofilm (zone 3), cell orientation shows increased deviation from the vertical alignment, though fluorescent cells still appear to be spatially aggregated (**Fig 1B**).

Growth at the liquid–air interface constitutes another popular model system that has provided insights into mechanisms of biofilm formation [46,51,76,77]. To examine whether striated arrangement, similar to that observed in the macrocolony biofilm, develops in a biofilm grown at the liquid–air interface (pellicle) we used the same ratio of marked and unmarked cells as the inoculum for pellicle biofilm cultures. After 3 days of growth, pellicle biofilms were transferred to agar-solidified media and immediately embedded in paraffin and thin-sectioned. Microscopic analysis revealed that mixing-assay pellicle biofilms also show fluorescent striations (**Fig 1D**). Each of the pellicles imaged showed striations covering most or all of the biofilm depth and containing cells in random orientations. These results are consistent with those of a separate study by our group, in which we have developed a technique for live, high-resolution imaging of cellular movement and the establishment of striations during pellicle development [78]. Similar to our observation, Puri and colleagues also recently found that *E. coli* pellicle biofilms inoculated from mixed suspensions of GFP- and mScarlet-expressing cells show striation formation [25]. Together, our findings demonstrate that clonal subpopulations of *P. aeruginosa* cells arrange in vertical striations perpendicular to the gel- or liquid–air interface during macrocolony or pellicle biofilm growth.

## Biofilm striations resolve and become longer over time

The presence of clonal striations in 3-day-old biofilms raised the question of how these features develop over the course of the incubation. To examine this, we grew replicate mixing-

assay biofilms and sacrificed the biofilms for microscopic analysis at a series of time points. Fluorescence imaging showed that PA14 cells in macrocolony biofilms are randomly oriented and distributed at 6 and 12 h of incubation (**Fig 1C**). At 24 h of incubation, the upper two-thirds of the biofilm began to show clustering and increased vertical orientation of fluorescent cells that gradually becomes more resolved over time. Between 24 and 72 h of incubation, both zone 1 and zone 2 showed gradual increases in height (3-fold and 2.6-fold, respectively). These observations suggest that cells in zone 1 are able to move away from each other, while in contrast cells in zone 2 remain positioned end-to-end, after division (**Fig 1C**).

## Resource availability affects the organization of cellular arrangement zones and profiles of metabolic activity in biofilms

Biofilms growing at interfaces form resource gradients as they get thicker due to diffusion limitation and consumption by cells closer to the boundary [1,79]. These gradients define chemical microniches that promote physiological differentiation. We hypothesized that changes in resource (in this case, tryptone) availability would alter cellular arrangement in *P. aeruginosa* macrocolonies. We tested this assumption by growing mixing-assay biofilms on different concentrations of tryptone and imaging thin sections by fluorescence microscopy. As expected, biofilms grew thicker when we provided increasing amounts of tryptone, indicating that a vertical tryptone gradient formed across biofilm depth with limiting concentrations at the air interface (**Fig 2B**). We were surprised to find that on 0.25% tryptone fluorescent striations and vertically oriented cells were visible in the zone close to the biofilm base, while dispersed and randomly oriented cells were visible in the upper portion of the biofilm. In contrast, biofilms grown on 0.5% tryptone showed an overall zone organization similar to that seen on 1% tryptone. The ordered region of these biofilms, however, appeared more similar to zone 3 of 1%-tryptone grown biofilms in that cellular orientation showed more deviation from vertical and striations were thicker and not as well-segregated as in biofilms grown on the higher concentration (**Fig 2B**). This further suggests that the level of dispersion displayed by cells in a given zone does not correlate with the absolute concentration of tryptone: for example, the disordered zone corresponded to a tryptone-limited region in the 0.25%-tryptone-grown biofilm, while the disordered zones instead corresponded to the most tryptone-replete regions in the 0.5%- and 1%-tryptone-grown biofilms. These observations show that striation formation is sensitive to nutrient availability but suggest that the absolute concentration of tryptone is not the sole factor determining the organization of cellular arrangement zones in biofilms.

Because gradients within biofilms affect physiological status, we next sought to test whether the effect of tryptone concentration on cellular arrangement zone properties and organization correlated with effects on metabolic activity. Our groups have established a protocol to visualize metabolic activity in biofilm thin sections using SRS microscopy [60]. In this method, biofilms are grown on agar plates, then transferred to a medium containing $D_2O$ for 24 h (**Fig 2A**) before the preparation of thin sections. Incorporation of D into D-C bonds is an indicator of metabolic activity and is detected by SRS microscopy. We found that, for biofilms grown on each concentration of tryptone, metabolic activity peaked at a depth of 40 to 70 μm (**Fig 2B**). To ensure that this activity pattern is not a result of uneven access to $D_2O$ in the biofilm, we quantified its distribution by SRS microscopy and found it to be uniform (**S2 Fig**). Microelectrode measurements have shown that *P. aeruginosa* biofilms grown at gel–air interfaces contain oxygen gradients [66,79]. The observation that maximal metabolic activity occurs at a consistent depth in biofilms grown at different tryptone concentrations suggests that maximum metabolic activity occurs where the relative concentrations of oxygen and tryptone are optimized [80]. Strikingly, the region with maximum metabolic activity overlapped with a

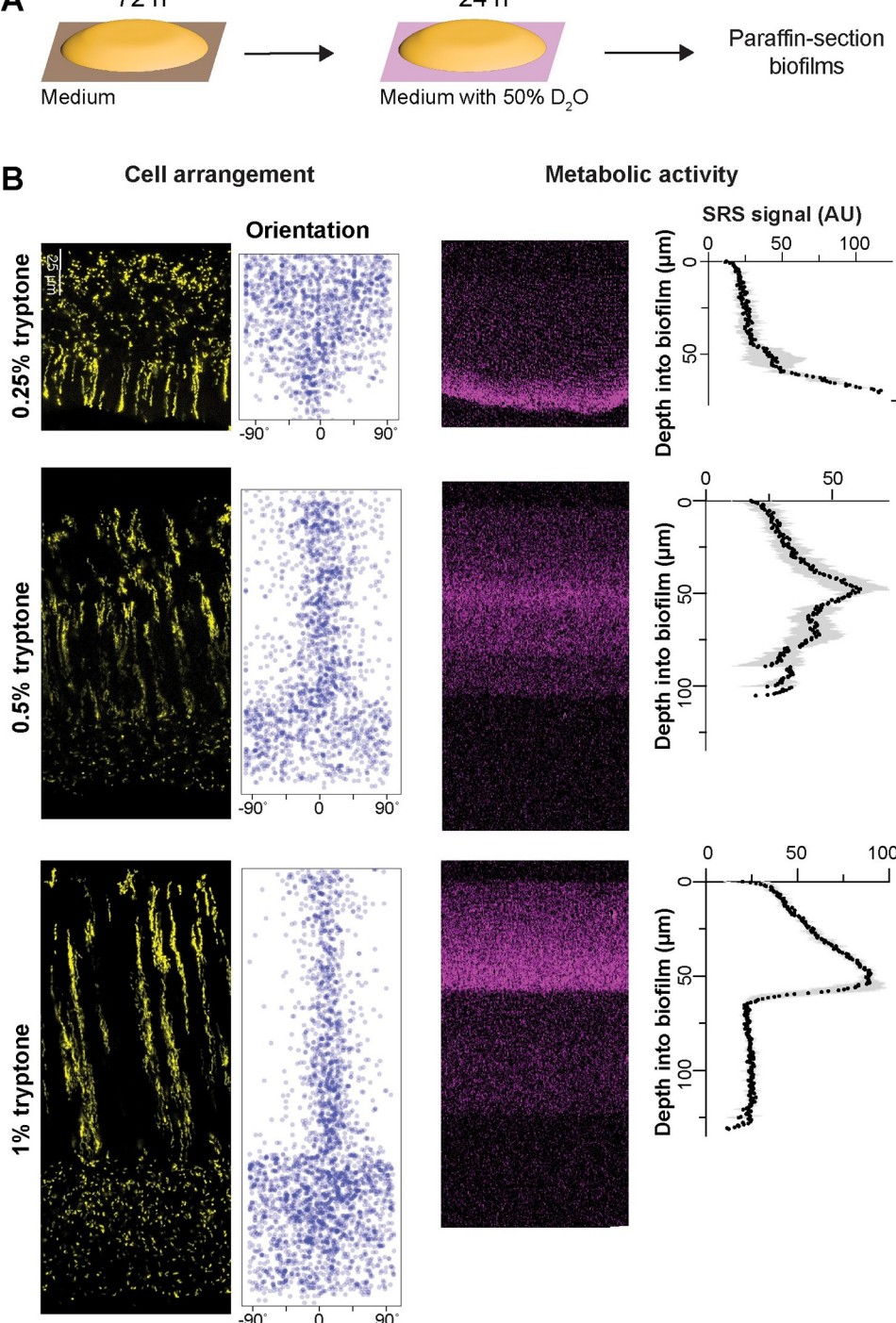

**Fig 2. Resource availability affects the organization of cellular arrangement zones and metabolic activity in biofilms.** (**A**) Experimental setup for growing *P. aeruginosa* biofilms on agar plates and their subsequent transfer to medium containing $D_2O$ for analysis of metabolic activity by SRS microscopy. (**B**) **Left**: Fluorescence micrographs and quantification of cellular orientation across depth for thin sections from mixing-assay biofilms grown on different concentrations of tryptone. Scale bar applies to all images. **Right**: SRS microscopy images and SRS signal across depth for thin sections from mixing-assay biofilms grown on different concentrations of tryptone. SRS signal represents the average of 3 biological replicates with shading indicating the standard deviation. The data underlying this figure can be found in S1 Data.

region containing vertically oriented cells in biofilms grown at all tryptone concentrations (**Fig 2B**). When macrocolonies are grown on solidified media [16,46], the electron donor (here, this is tryptone, which is also the carbon source) and acceptor (here, oxygen) are typically provided on opposite sides of the biofilm. To assess whether reversing the electron-acceptor gradient would affect striation formation, we grew the biofilm anaerobically by adding potassium nitrate to the growth medium [81]. Under this condition, biofilms did not exhibit striations, underscoring the critical role of the oxygen gradient in their formation. Accordingly, biofilms cultured aerobically on nitrate-enriched medium displayed striations similar to those observed in the absence of added nitrate, suggesting that nitrate itself does not influence striation formation (**S3 Fig**).

### Global regulators and cell surface components affect cell patterning

*P. aeruginosa* colony biofilm structure at the macroscale is determined primarily by the regulation of exopolysaccharide production in response to environmental cues and cellular signals [46,53,82–84]. To uncover the genetic determinants of biofilm cellular arrangement at the microscale, we conducted a targeted screen of 48 mutants lacking regulatory proteins, cell surface components, and other factors that contribute to WT biofilm formation and physiology in macrocolonies and other biofilm models (**Fig 3A**) [83–86]. For each of these mutants, a corresponding, constitutively fluorescent strain was generated that expressed the $P_{A1/04/03}$-*mScarlet* construct in the mutant background. Mixed biofilms of each mutant were grown for 3 days and thin sections were prepared for microscopic analysis.

Imaging allowed us to identify 4 classes of cellular arrangement phenotypes, which we refer to as "striated," "bundled," "disordered," or "clustered" (**Fig 3B**). Thirty-seven out of the 48 screened mutants showed the "striated" cellular arrangement phenotype of the WT (**Fig 3A and 3B**). The remaining 11 mutants showed cellular arrangement phenotypes that differed from the WT. The "bundled" phenotype was observed in all tested genetic knockouts defective in the production and function of the type IV pilus (i.e., *pil* gene mutants and the mutant lacking RpoN, which is required for *pilA* expression) [87,88] (**Figs 3A, 3B, and S5**). In this phenotype, cells are oriented as they are in WT striations, i.e., perpendicular to the air interface. However, the bundles are wider than WT striations and spacing between bundles is broader than it is between WT striations. We hypothesize that pilus-mediated motility, known as twitching, serves to distribute cells in the x-y plane before vertical growth in zone 2 promotes the formation of segregated striations in the WT. According to this model, bundled striations form in twitching-defective mutants because daughter cells are less able to move away from their relatives before the elongation of zone 2.

Biofilms with the "disordered" phenotype display a "zone 1"-type distribution of cells across their full depth, i.e., cells are evenly spaced and randomly oriented (**Figs 3B and S5**). This was observed in mutants with disruptions in the GacS/GacA two-component system or in LasR, both of which constitute regulators that affect the expression of hundreds of genes. In addition, deletion of the O-antigen biosynthetic gene *wbpM* gave rise to this phenotype (**Fig 3B**). O-antigen is the outermost portion of lipopolysaccharide (LPS), a major constituent of the outer membrane of gram-negative bacteria [89]. The appearance of these biofilms suggests that in these mutants, related cells are fully dispersed from each other. Finally, Δ*ssg* and Δ*wapR*–mutants that are predicted to form LPS without the O-antigen attached (**Figs 3B and S5**) [90–92]—gave rise to a "clustered" phenotype. These biofilms showed random cell orientation, but fluorescent cells also formed clusters with vertical elongation in zones 2 and 3, indicating that these mutants retained some ability to restrict movement of related cells in the x-y plane (**Figs 3B and S5**). Our identification of mutants with defects in O-antigen attachment in the cellular

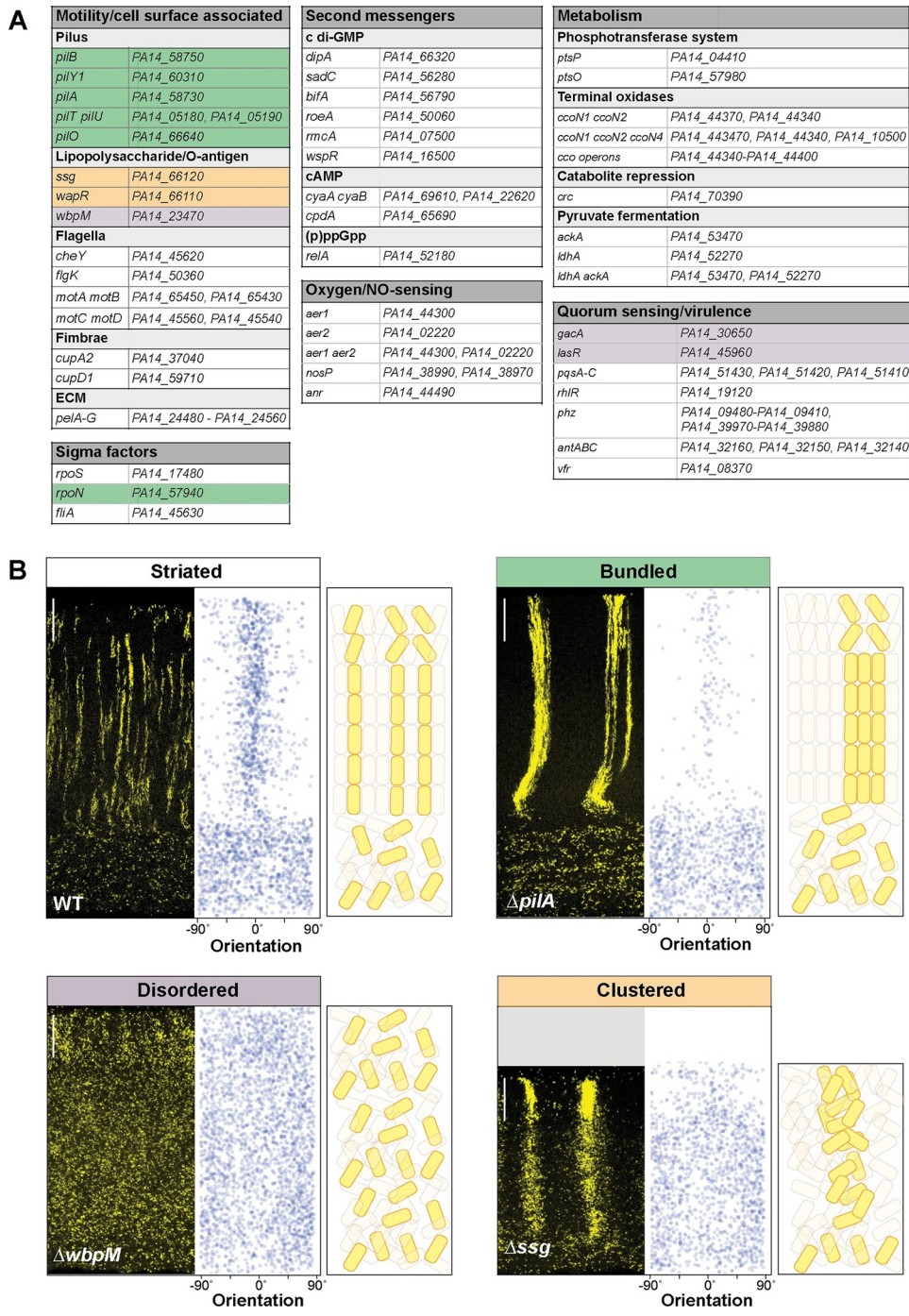

**A**

| Motility/cell surface associated | |
|---|---|
| **Pilus** | |
| pilB | PA14_58750 |
| pilY1 | PA14_60310 |
| pilA | PA14_58730 |
| pilT pilU | PA14_05180, PA14_05190 |
| pilO | PA14_66640 |
| **Lipopolysaccharide/O-antigen** | |
| ssg | PA14_66120 |
| wapR | PA14_66110 |
| wbpM | PA14_23470 |
| **Flagella** | |
| cheY | PA14_45620 |
| flgK | PA14_50360 |
| motA motB | PA14_65450, PA14_65430 |
| motC motD | PA14_45560, PA14_45540 |
| **Fimbrae** | |
| cupA2 | PA14_37040 |
| cupD1 | PA14_59710 |
| **ECM** | |
| pelA-G | PA14_24480 - PA14_24560 |

| Sigma factors | |
|---|---|
| rpoS | PA14_17480 |
| rpoN | PA14_57940 |
| fliA | PA14_45630 |

| Second messengers | |
|---|---|
| **c di-GMP** | |
| dipA | PA14_66320 |
| sadC | PA14_56280 |
| bifA | PA14_56790 |
| roeA | PA14_50060 |
| rmcA | PA14_07500 |
| wspR | PA14_16500 |
| **cAMP** | |
| cyaA cyaB | PA14_69610, PA14_22620 |
| cpdA | PA14_65690 |
| **(p)ppGpp** | |
| relA | PA14_52180 |

| Oxygen/NO-sensing | |
|---|---|
| aer1 | PA14_44300 |
| aer2 | PA14_02220 |
| aer1 aer2 | PA14_44300, PA14_02220 |
| nosP | PA14_38990, PA14_38970 |
| anr | PA14_44490 |

| Metabolism | |
|---|---|
| **Phosphotransferase system** | |
| ptsP | PA14_04410 |
| ptsO | PA14_57980 |
| **Terminal oxidases** | |
| ccoN1 ccoN2 | PA14_44370, PA14_44340 |
| ccoN1 ccoN2 ccoN4 | PA14_443470, PA14_44340, PA14_10500 |
| cco operons | PA14_44340-PA14_44400 |
| **Catabolite repression** | |
| crc | PA14_70390 |
| **Pyruvate fermentation** | |
| ackA | PA14_53470 |
| ldhA | PA14_52270 |
| ldhA ackA | PA14_53470, PA14_52270 |

| Quorum sensing/virulence | |
|---|---|
| gacA | PA14_30650 |
| lasR | PA14_45960 |
| pqsA-C | PA14_51430, PA14_51420, PA14_51410 |
| rhlR | PA14_19120 |
| phz | PA14_09480-PA14_09410, PA14_39970-PA14_39880 |
| antABC | PA14_32160, PA14_32150, PA14_32140 |
| vfr | PA14_08370 |

**Fig 3. Specific global regulators and cell surface components are required for WT cell patterning in colony biofilms.** (**A**) List of genes mutated and then screened for altered cellular arrangement across depth in biofilms. Those showing altered cellular arrangement are shaded and colors correspond to the phenotype categories shown in (B). (**B**) Fluorescence micrographs of thin sections from WT and indicated mutant biofilms grown on 1% tryptone + 1% agar for 3 days. Biofilm inocula contained 2.5% cells that constitutively express mScarlet. mScarlet fluorescence is colored yellow. Quantification of orientation across depth is shown for each image, and cartoons of cellular arrangement are shown for each phenotype category. Images shown are representative of at least 2 independent experiments. Scale bars are 25 μm. The data underlying this figure can be found in S1 Data.

arrangement screen is interesting in light of the fact that O-antigen has also been implicated in the formation of "stacked" aggregates in cultures grown using an in vitro cystic fibrosis model [93]. In this model, the aggregates formed by WT *P. aeruginosa* contain cells that are packed lengthwise while those formed by Δ*ssg* and Δ*wbpM* mutants appear as "disorganized clumps," with Δ*ssg* forming larger clumps than Δ*wbpM*. We note that our screen did not yield any mutants that exhibit vertical cellular orientation without clustering of related cells (or striation formation). This may indicate that, in zone 2, dispersion after cell division overrides any ability to establish vertical orientation.

The "striated" (i.e., WT-like) class contains several categories of mutants that show altered biofilm development in diverse models [83,94]. These include mutants with altered levels of second messengers; those with defects in primary metabolism; those with deletions of oxygen-sensing or NO-sensing proteins; and those with defects in the production of flagellar components, fimbriae, Pel polysaccharide, or selected virulence factors. Some specific examples are provided by the following mutants, which all exhibited WT cellular arrangement in the mixing assay, but show altered development in a colony morphology assay: Δ*rmcA* (hyperwrinkled macrocolony with increased spreading relative to WT) [53], Δ*flgK* (hyperwrinkled macrocolony) [95], and Δ*pel* (smooth macrocolony) [46,62]). We grew the WT and mutants with altered cellular arrangement phenotypes in the colony morphology assay and found that they exhibited diverse morphologies. Mutants showing the bundled phenotype formed biofilms that were more compact and smoother than the WT. The Δ*ssg* mutant, which shows a clustered phenotype, was also compact and smooth. Mutants that display a disordered phenotype, however, did not produce a consistent colony morphology, with morphotypes ranging from compact and "dimpled" to WT-like to smooth (**S4 Fig**). Together, these findings show that effects on biofilm formation and development at the macroscale do not correlate with effects on cellular arrangement at the microscale and across depth (**S4 Fig**).

## A retractable pilus is required for WT cellular arrangement across biofilm depth

To gain further insight into the structure of the bundled phenotype, which arises in mutants with pilus-related defects (**Figs 3A, 3B, 4A, and S5**), we used high-resolution light sheet microscopy to take top-view images. For this method, the medium was inoculated using a cell suspension containing the $P_{A1/04/03}$-mScarlet strain at 1% and nonfluorescent cells at 99%. Macrocolonies were imaged after 19 h of growth. The WT and Δ*pilA* biofilms showed patterns of fluorescence consistent with the distribution of fluorescent cells we observe in biofilm thin sections taken between 12 and 24 h (**Figs 1C and 3B**): striations appear as puncta, or as vertical columns that are tilted at an angle away from perfect perpendicularity with the biofilm–air interface. Fluorescent striations in the Δ*pilA* biofilms are thicker and occur at greater distances away from each other than those in WT biofilms. In Δ*pilA*, the striations also generally show a vertical orientation in the biofilm center, but diagonal orientations that radiate toward the colony edge in the biofilm outgrowth (**Fig 4A**). This arrangement is reminiscent of the "hedgehog"-like cellular ordering that has been reported for *V. cholerae* biofilms grown from single cells under flow [19,20].

The phenotypes of the individual *pil* mutants allow us to make inferences about the contributions of the pilus and of twitching motility to cellular arrangement in WT biofilms. Twitching motility is conferred via repeated pilus extension, attachment, and retraction. The primary constituent of the pilus is PilA, also known as pilin. PilB is the extension motor—responsible for pilus assembly—while PilTU is the retraction motor [96,97]. PilO and PilY1 are also both required for pilus assembly; moreover, recent studies have revealed roles for both PilY1 and

**Fig 4. A functional pilus is required for WT cellular arrangement.** (**A**) High-resolution light sheet microscopy images of 19-hour-old biofilms, with 1% of cells expressing fluorescent protein (colored yellow). (**B**) Fluorescence micrograph and quantification of orientation for WT and each of the indicated mutants grown in the mixing assay and thin-sectioned. Scale bars are 25 μm. The extent of pilus function present in each strain is indicated by a cartoon. The data underlying this figure can be found in S1 Data.

PilT in surface sensing [98–100]. All of the *pil* mutants we tested show bundled striations in zone 2 highlighting that a functional, i.e., extendable and retractable, pilus is required for the formation of WT striations (**Figs 4B** and **S5**); however, Δ*pilTU* is unique in that it also shows bundled striations in zone 1 (**Fig 4B**).

## PilA is required for the zone 2 dispersion observed in O-antigen synthesis and attachment mutants

The results of our screen provide clues regarding the molecular determinants of cellular arrangement in *P. aeruginosa* macrocolonies. First, they suggest that a functional pilus is required for cellular distribution specifically and immediately before the formation of zones 2/3, i.e., that related cells twitch away from each other before they divide and form vertical lineages. However, the pilus is not required for cellular dispersion in zone 1, vertical striation formation, or vertical orientation (**Figs 3B, 4A and 4B**). Second, they suggest that modifications to the structure of LPS (**Fig 5A**) can affect both cellular orientation and the clustering of related cells (or striation formation). These inferences raise the question of whether these cell surface components show interactions at the regulatory or phenotypic level. We tested this by evaluating PilA protein levels in O-antigen synthesis/attachment mutants and by examining cellular arrangement in combinatorial mutants.

To assess the effect of LPS-related mutations on cellular PilA levels, we grew the WT, Δ*wbpM*, Δ*wapR*, and Δ*ssg* strains as macrocolony biofilms for 3 days before preparing whole-cell and sheared whole-cell lysates. PilA levels were comparable for sheared whole-cell (**Figs**

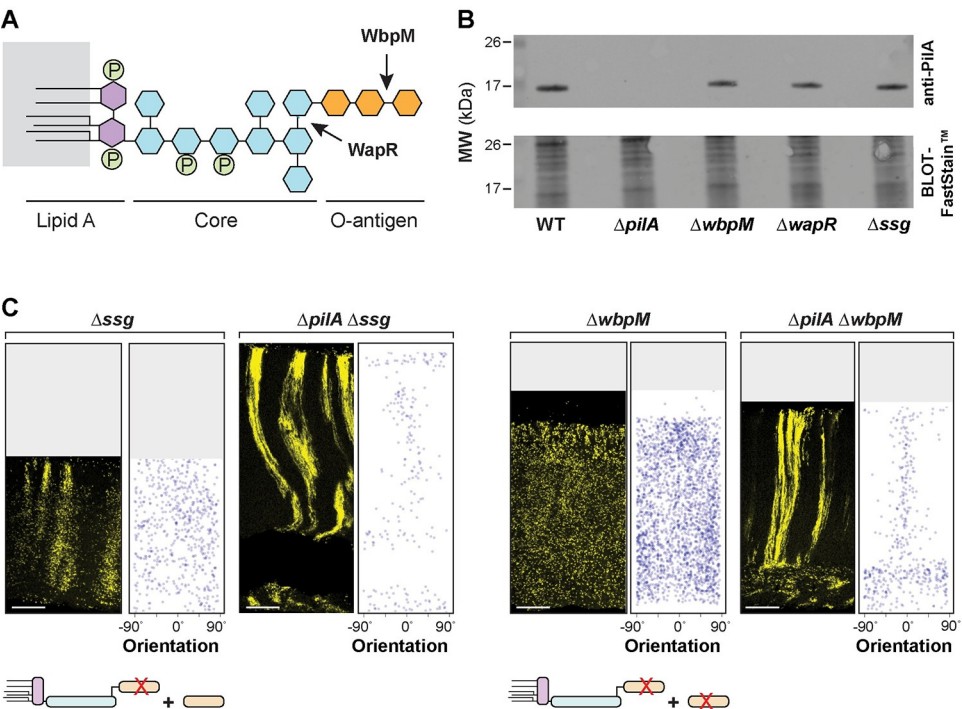

**Fig 5. PilA is required for the disordered cellular arrangement phenotypes of O-antigen mutants.** (**A**) Schematic of LPS and O-antigen indicating the bonds affected by WapR and WbpM activity. Hexagons represent monosaccharides and circles represent phosphate groups. The major components of LPS are color-coded. (**B**) Western blot showing PilA protein levels in macrocolony biofilms of WT and the indicated mutants. Equal amounts of total protein from sheared whole-cell lysates for each strain were resolved by SDS-PAGE using a 15% polyacrylamide gel. The PilA protein was detected using an anti-PilA antibody. (**C**) Fluorescence micrograph and quantification of orientation for each of the indicated mutants grown in the mixing assay and thin-sectioned. A cartoon representation of LPS and O-antigen (colors corresponding to panel (**A**)) indicates whether unattached and/or attached O-antigen are present in each strain. Images shown are representative of at least 2 independent experiments and mScarlet fluorescence is colored yellow. Scale bars are 25 μm. The data underlying this figure can be found in S1 Data.

5B and S6A) and whole-cell (S6B and S6C Fig) lysates in all tested strains. These findings indicate that the distinct cell-arrangement phenotypes of mutants with defects in LPS synthesis or attachment are not due to differences in total PilA levels or PilA extrusion.

To examine phenotypic interactions between mutations that affect O-antigen synthesis/attachment and a mutation that affects pilin production, we created double mutants lacking combinations of the genes *wbpM*, *ssg*, and *pilA*, both with and without the P$_{A1/04/03}$-*mScarlet* construct. Macrocolony biofilm mixing assays revealed that Δ*wbpM* is epistatic on Δ*ssg* (S5 Fig). This indicates that the presence of O-antigen that is not attached to LPS in Δ*ssg* is sufficient to promote the clustering of related cells observed in this single-gene mutant. Intriguingly, they also revealed that Δ*pilA* is epistatic on Δ*wbpM* and Δ*ssg*: deleting *pilA* in these backgrounds restored the formation of (bundled) striations (Fig 5C). These observations suggest that *pilA* promotes dispersion in both the striated WT and the non-striated mutants, such as Δ*ssg* and Δ*wbpM*. In the absence of O-antigen-modified LPS, *pilA* also appears to promote random cellular orientation. This finding highlights the intricate interplay between pilus function, O-antigen synthesis/attachment, and cellular organization in the construction of *P. aeruginosa* biofilms. Interestingly, although it remains unconfirmed in *P. aeruginosa* PA14, pilus glycosylation with O-antigen has been described in other *P. aeruginosa* strains [101].

## Cell patterning influences the distribution of substrates across biofilm depth

The distribution of resources throughout a biofilm can be affected by a variety of factors, including its physical structure, properties of the matrix, and the metabolic activity of the cells within [30,31]. We assessed the effects of cellular arrangement on uptake and distribution using substrates that are vastly different in size. To examine distribution of a small substrate within *P. aeruginosa* macrocolonies, we engineered our strains of interest to contain the *rhaSR*-P*rhaBAD* inducible promoter system, designed to express *mScarlet* in response to the small molecule L-rhamnose [102] (**Fig 6C**). (Rhamnose has a topological polar surface area of 90.2 $Å^2$.) Corresponding strains that constitutively produce eGFP were made so that related-cell distribution and rhamnose-induced expression could be examined in the same thin section. Macrocolony biofilms were grown from inocula that contained the constitutive eGFP-producing strain at 2.5% and the RhaSR-P*rhaBAD*-controlled mScarlet-producing strain at 97.5%. These were incubated for 3 days on our standard medium, then transferred to medium containing L-rhamnose for 5 h before they were embedded in paraffin and prepared for thin sectioning (**Fig 6A**). Using this system, we found that in zones 1 and 2, mScarlet production was undetectable or only slightly above that observed in uninduced controls for all strains, indicating that pleiotropic effects inhibited activity of the *rhaSR*-P*rhaBAD* inducible promoter system in this part of the biofilm. However, all strains showed production of mScarlet in zone 3, i.e., at the biofilm–air interface and the relative fluorescence of this zone can therefore be compared. WT, Δ*pilA*, and Δ*pilA*Δ*wbpM* biofilms showed similar, moderate levels of mScarlet production in this region. They also showed heterogeneous and often vertically arranged clustering of enhanced mScarlet production, which may arise from gaps or channels in the biofilm structure that allow for enhanced penetration of molecules such as rhamnose from the growth medium into the biofilm (**Figs 6B, 6D,** and **S7**). In the 2 disordered mutants Δ*wbpM* and Δ*gacA* (**Figs 6B, 6D,** and **S7**), mScarlet production in zone 3 was greatly enhanced. In contrast, in planktonic culture we did not observe differences between WT and these mutants with respect to rhamnose-dependent fluorescence (**S8 Fig**) [103]. These findings suggest that lengthwise cellular packing and/or vertical cellular orientation affect the transport of nutrients across biofilm depth. We note that fluorescence produced by the rhamnose-inducible reporter system depends not only on the distribution of this sugar but also its uptake into the cells and subsequent expression of the reporter.

To more directly assess the diffusion of a small molecule through biofilms, we live-imaged the distribution of the water-soluble and fluorescent dye Sulfo-Cy5 NHS-ester. (Sulfo-Cy5 NHS-ester has a topological polar surface area of 261 $Å^2$.) We compared three-day-old pellicle biofilms formed by *P. aeruginosa* WT and Δ*wbpM* using light sheet microscopy [78] and observed strain-specific differences in dye distribution (**S9 Fig**). Although reporter expression and dye distribution were differentially affected by the *wbpM* mutation, highlighting that multiple factors contribute to these fluorescence patterns, the dye distribution phenotype nevertheless constitutes further evidence suggesting that cell arrangement affects substrate distribution.

To further examine the effect of cellular arrangement on transport within biofilms, we tested the uptake and distribution of fluorescent microspheres (diameter: 200 nm). Microspheres were spread on the surface of an agar plate before inoculation for macrocolony biofilm growth. After a three-day incubation, the biofilms were imaged from the top or cryosectioned and imaged immediately (**Fig 6E and 6G**). While microspheres were visible at the air-exposed surface of WT and Δ*wbpM* biofilms, few or no microspheres were detected at the surface of biofilms formed by mutants lacking type IV pili (Δ*pilA* and Δ*pilA*Δ*wbpM*) (**Fig 6F**), suggesting

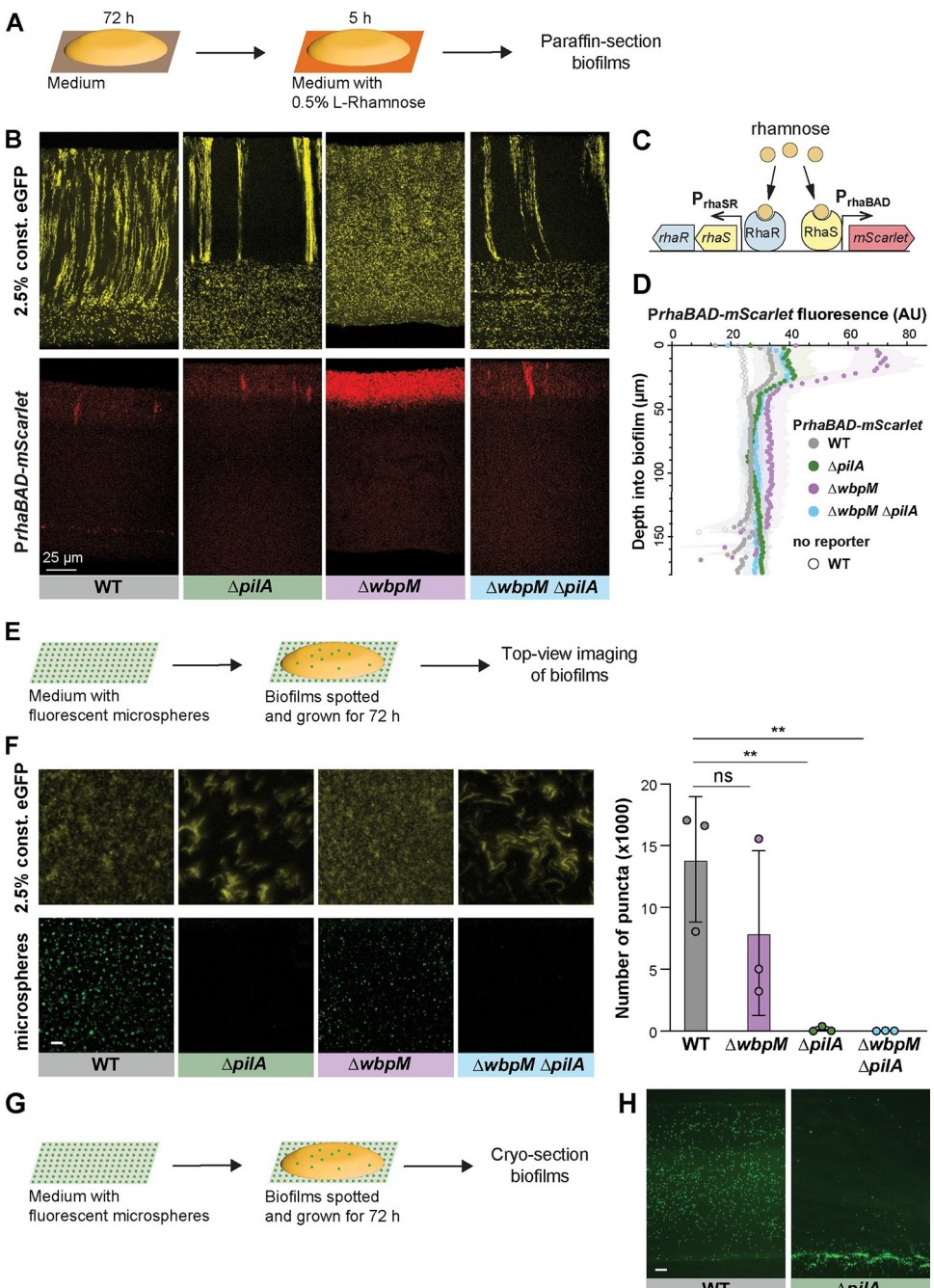

**Fig 6. Cellular arrangement affects the uptake of substrates into colony biofilms.** (**A**) Schematic illustration of the experimental setup for growing *P. aeruginosa* biofilms on agar plates and their subsequent transfer to medium containing L-rhamnose. (**B**) Fluorescence micrographs of thin sections from WT and indicated mutant biofilms. Biofilm inocula contained 2.5% cells that constitutively express eGFP and 97.5% RhaSR-P*rhaBAD*-controlled mScarlet-producing strain. Top panels show eGFP fluorescence (colored yellow) and bottom panels show the mScarlet fluorescence for each thin section. Scale bar applies to all images. (**C**) Schematic of RhaSR-P*rhaBAD* expression system driving mScarlet production. (**D**) Quantification of mScarlet fluorescence shown in (B). Shading represents standard deviation of biological triplicates. (**E**) Schematic of the experimental setup for growing *P. aeruginosa* macrocolony biofilms on agar plates with fluorescent microspheres (200 nm). (**F**) **Left**: Top-view images taken from center of macrocolony biofilms that were grown on medium with microspheres (colored green). Biofilms contained 2.5% cells constitutively expressing eGFP (colored yellow). Scale bar is 25 μm and applies to all images. **Right**: Quantification of microspheres visible in top-view images. Each data point is a biological replicate, bar height indicates the mean of these replicates; *p* values are based on two-sided unpaired *t* tests (n.s., not significant; ** *p* ≤ 0.01). (**G**) Schematic illustration

of the experimental setup for growing biofilms on agar plates with fluorescent microspheres for cryosectioning. (**H**) Fluorescence micrographs of cryosections of biofilms grown on microspheres (colored green). Scale bar is 25 μm and applies to both images. The data underlying Fig 6D and 6F can be found in S1 Data.

that the pilus affects biofilm microanatomy in a manner that facilitates particle uptake. In cryo-sections, we found that, indeed, microspheres were evenly distributed within WT biofilms, while for Δ*pilA* they were barely able to enter the biofilm at the agar interface (**Fig 6H**). This observation indicates that cells in Δ*pilA* biofilms are more tightly packed than those in WT biofilms and that this inhibits the movement of particles. In summary, the distribution of sub-strates of different sizes (rhamnose, dye, and microspheres)—into and within biofilms—was affected by changes in cell arrangement.

## Altered cellular arrangement phenotypes correlate with effects on the distributions of metabolically active and surviving cells

The correlations we observed between cellular arrangement and metabolic activity (**Fig 2B**), and between cellular arrangement phenotypes (**Fig 3B**) and substrate distribution (**Figs 6B, 6F, 6H, and S7**), in biofilms suggested that mutations that affect microorganization across depth could affect physiological patterning. We therefore tested whether mutant biofilms with altered cellular arrangement show altered distributions of metabolic activity (**Figs 7A and S10**). SRS imaging of mutant biofilm thin sections indeed revealed that the band of metabolic activity is shifted closer to the top of the biofilm in Δ*wbpM* (peak "2") when compared to WT (peak "3") and is unchanged in the Δ*pilA* mutant. The metabolic activity profile of Δ*wbpM* also shows a "shoulder" of increased activity (peak "1") that is not present in the WT and Δ*pilA*. A double deletion of *wbpM* and *pilA* lacked this shoulder (**Fig 7B**).

To further interrogate physiological status, we assessed cell death across depth in biofilms by exposing them to propidium iodide (PI), a DNA stain that is excluded from bacteria with intact membranes. For this assay, three-day-old biofilms were transferred to plates containing medium supplemented with 50 μM PI and incubated for 5 h before sample preparation and thin-sectioning (**Fig 7C**). Microscopic imaging revealed increased PI staining in thin sections of Δ*wbpM* biofilms, particularly in the upper region of the biofilm, when compared to WT. Thin sections of WT and Δ*pilA* biofilms showed similar levels and distributions of PI staining (**Fig 7D**).

Together, the results of our metabolic activity and cellular survival profiling experiments are consistent with the notion that more substrates from the growth medium reach the biofilm region at the air interface in the Δ*wbpM* mutant. The altered cellular arrangement in the Δ*wbpM* mutant biofilm appears to disrupt the normal stratification of metabolic activity, push-ing it closer to the biofilm surface (**Fig 7B**). The increased PI staining in this region could be a consequence of this accelerated metabolic activity, leading to quicker exhaustion of local resources (i.e., tryptone) and, thus, a higher incidence of nonviable cells.

## Altered cellular arrangement affects antibiotic tolerance

The physiological status of a bacterium affects its susceptibility to antibiotics, and particularly those that target biosynthetic reactions and/or that require functional membrane transport processes for uptake [104–108]. Because we had observed that mutants with altered cellular arrangements showed distinct patterns of substrate distribution and metabolic activity within biofilms (**Figs 6B, 6F, 6H, 7B, S7, and S10**), we hypothesized that they would be differentially susceptible to antibiotic treatment. To test this, we transferred three-day-old *P. aeruginosa*

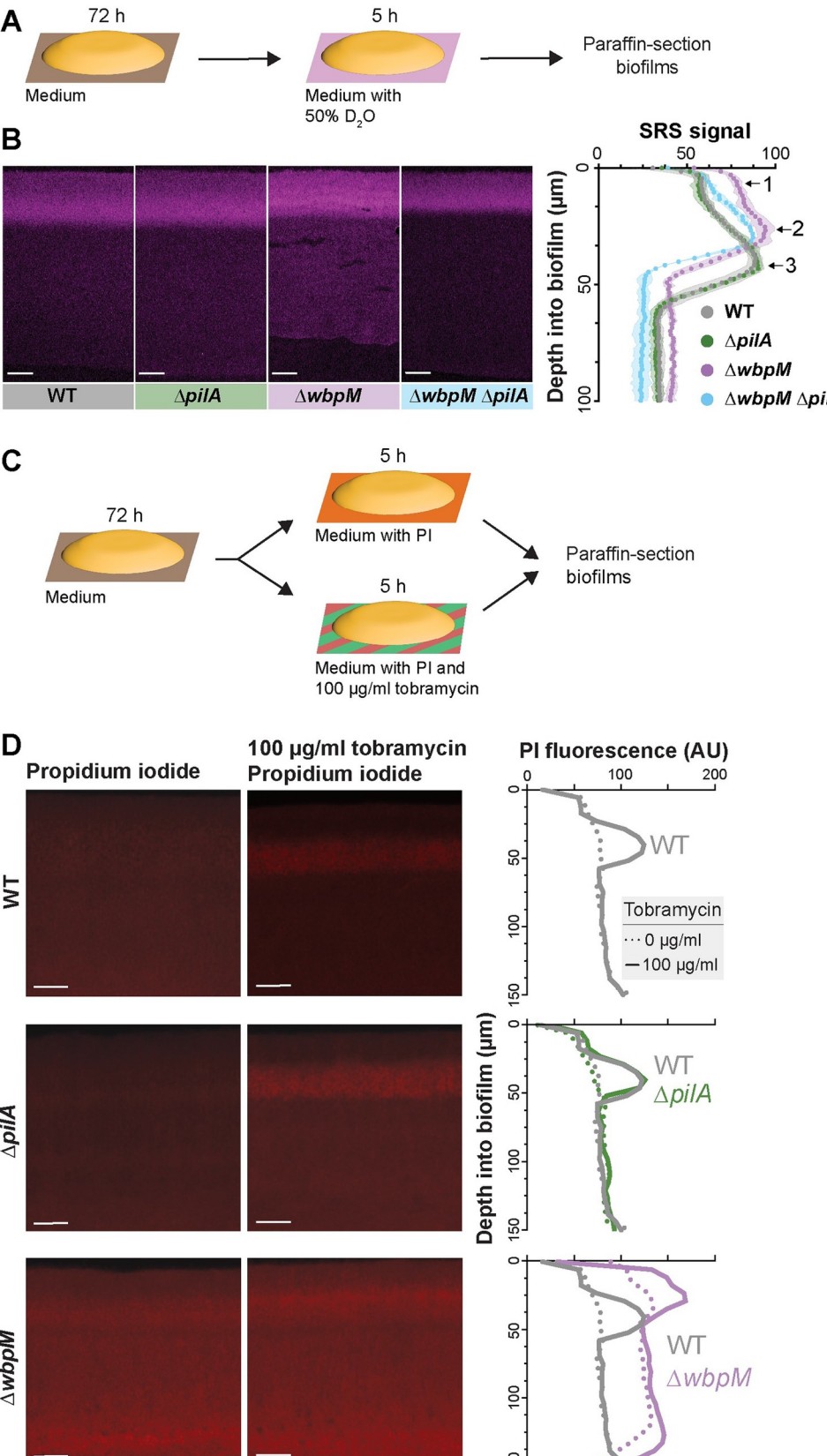

**Fig 7. Mutations affecting cellular arrangement alter metabolic activity and antibiotic tolerance profiles within biofilms.** (**A**) Schematic illustration of the experimental setup for growing biofilms on agar plates and their subsequent transfer to a medium containing $D_2O$ for analysis of metabolic activity using SRS microscopy. (**B**) **Left**: SRS micrographs of WT and mutant biofilm thin sections. **Right**: Graph showing the quantitative shift in metabolic activity distribution in different mutants. (**C**) Schematic illustration of the experimental setup for growing biofilms on agar plates and their subsequent transfer to a medium containing propidium iodide or propidium iodide + tobramycin (**D**) **Left**: Fluorescence micrographs of three-day-old biofilms exposed to PI. **Center**: Fluorescence micrographs of biofilms treated with tobramycin and PI. **Right**: Graphs showing quantification of PI staining. Images are representative of at least 2 independent experiments. (B, D) Scale bars are 25 μm. The data underlying Fig 7B and 7D can be found in S1 Data.

macrocolonies to a medium containing PI and tobramycin, an aminoglycoside antibiotic that requires a transmembrane proton motive force for its bactericidal action, and incubated for 5 h before biofilms were prepared for thin-sectioning [109]. We infer that an increase in PI staining indicates a decrease in antibiotic tolerance [110,111]. Comparison to thin sections from biofilms that had not been treated with tobramycin (**Fig 7D**) revealed that, in WT and Δ*pilA*, the antibiotic killed cells in the region spanning from 20 to 60 μm from the air–biofilm interface, which corresponded to the regions of high metabolic activity (**Fig 7B**). This observation is consistent with prior work in which alternative methods for detecting metabolic activity had indicated that physiological status impacts antibiotic tolerance in biofilms [58,106,112]. In contrast to those obtained for WT and Δ*pilA*, biofilm thin sections for Δ*wbpM* and Δ*gacA* showed tobramycin-dependent cell death in a region that was closer to the air–biofilm interface (0 to 40 μm depth) (**Figs 7D, S11B and S11C**). This change corresponded to the shifted region of high metabolic activity that we had observed for Δ*wbpM* (**Fig 7B**) and the increased response to rhamnose at the biofilm–air interface for Δ*wbpM* (**Fig 6B**) and Δ*gacA* (**S7 Fig**) (we note that the *wbpM* mutation also has a more general negative effect on survival in biofilms as indicated by the overall increase in PI staining of untreated biofilms). In agreement with the dominant effect of the *pilA* deletion on cell arrangement (**Figs 5C and S11A**) and metabolic activity (**Fig 7B**), the peak of PI staining shifted back to a depth of 20 to 60 μm in the Δ*gacA*Δ*pilA* double mutant (**S11C Fig**). While cells are randomly oriented in Δ*wbpM* and Δ*gacA* biofilms, the deletion of *pilA* in these backgrounds results in vertical cellular arrangement as observed in WT and Δ*pilA* (**Figs 5C and S11A**). These patterns further highlight the correlation between cellular arrangement and antibiotic tolerance. They are also in line with prior work demonstrating that laterally aligned *P. aeruginosa* cells that have been aggregated via a depletion mechanism (e.g., via exposure to polymers that occur naturally in infection sites) show increased antibiotic tolerance [113].

Because our data suggest that cellular arrangement affects the transport of molecules across biofilm depth, two phenomena may be contributing to the differential impact of tobramycin treatment on ordered (WT) and disordered-mutant biofilms. First, tobramycin, like rhamnose, may move more readily through disordered biofilms than through ordered ones, leading to a shift in the zone of tobramycin-dependent cell death toward the biofilm–air interface. Alternatively or in addition, the greater availability of tryptone near the biofilm–air interface in disordered mutants like Δ*wbpM*—which appears to enhance metabolic activity in this zone—may also contribute to tobramycin susceptibility. Regardless of whether one or both of these phenomena are responsible for changes in the disordered-mutant profiles, our data indicate a relationship between cellular organization and the physiological properties of the biofilm that impacts antibiotic tolerance.

## Concluding remarks

This study has uncovered genetic and structural determinants that orchestrate cell arrangement within *P. aeruginosa* biofilms. A screen of 48 mutants representing genes implicated in

biofilm development and physiology identified 11 that impact cellular organization across depth, underscoring that this aspect of biofilm architecture is uniquely controlled. Our work also revealed correlations between cellular arrangement phenotypes and impacts on substrate distribution, metabolic activity, and antibiotic tolerance highlighting the critical role of micro-organization in determining metabolic status. An understanding of the relationships between microanatomy and physiology deepens our knowledge of bacterial multicellularity and provides avenues for the development of new therapeutic strategies. It also prompts further exploration into the potential parallels and divergences between biofilm mechanisms and multicellular organization in other organisms, paving the way for interdisciplinary insights.

## Materials and methods

### Bacterial strains and culture conditions

Bacterial strains used in this study are listed in **S1 Table**. Liquid cultures were routinely grown in lysogeny broth (LB) [114] at 37°C with shaking at 250 rpm.

### Construction of mutant *P. aeruginosa* strains

To create markerless deletion mutants in *P. aeruginosa* PA14, ~1 kb of flanking sequence from each side of the target gene were amplified using the primers listed in **S3 Table** and inserted into pMQ30 through gap repair cloning in *Saccharomyces cerevisiae* InvSc1 [115]. Each plasmid listed in **S2 Table** was transformed into *E. coli* strain UQ950, verified by Sanger sequencing, and moved into PA14 using biparental conjugation. PA14 single recombinants were selected on LB agar plates containing 100 μg/ml gentamicin. Double recombinants (markerless deletions) were selected on LB without NaCl and modified to contain 10% sucrose. Genotypes of deletion mutants were confirmed by PCR. Combinatorial mutants were constructed by using single mutants as hosts for biparental conjugation.

### Scanning electron microscopy of macrocolonies

Overnight precultures were diluted 1:100 in LB and grown to mid-exponential phase (OD at 500 nm ~ 0.5). OD values at 500 nm were read in a Spectronic 20D+ spectrophotometer (Thermo Fisher Scientific [Waltham, Massachusetts, United States of America]) and cultures were adjusted to the same OD. Five microliters of culture were spotted onto a bilayer plate (described below in "Colony biofilm mixing assay"). After 3 days of incubation, macrocolonies were carefully overlaid with 1% agar to preserve their structure. Blocks were then cut out, and the bottom layer of the agar was carefully removed, leaving an equal volume on top and below the macrocolony. The blocks were placed on filter paper when necessary to facilitate moving them with the least amount of disturbance, then placed into an empty glass Petri dish. The blocks were then incubated in McDowell Trump Fixative for 16 h at room temperature to allow fixation to occur. They were washed twice with 0.1 M PBS before being subjected to 1% (wt/vol) Osmium Tetroxide in 0.1 M PBS to further fix the samples. The samples were then washed twice with distilled water before being subjected to dehydration in an increasing graded ethanol series (35%, 50%, 75%, 2× 90%, 3× 100%). The blocks were washed twice in HMDS before being mounted onto aluminum stubs using double-sided round carbon stickers and placed into a desiccator. After air-drying thoroughly, the blocks were cut in half in order to visualize the internal structure and orientation of cells within the biofilm. The samples were then coated in gold using a Leica EM ACE600 Coater. The samples were visualized using a Helios Nanolab DualBeam 660 (FEI) operating at an accelerating voltage of 5 kV under a high vacuum.

## Colony biofilm mixing assay

Overnight precultures were diluted 1:100 in LB and grown to mid-exponential phase (OD at 500 nm ~ 0.5). OD values at 500 nm were read in a Spectronic 20D+ spectrophotometer (Thermo Fisher Scientific) and cultures were adjusted to the same OD. Adjusted cultures were then mixed in a 2.5:97.5 ratio of fluorescent:non-fluorescent cells. Five microliters of mixture were spotted onto a bilayer plate of 45 ml (bottom layer) and 15 ml (top layer) of 1% tryptone, 1% agar (Teknova [Hollister, California, USA] A7777) in a 10 cm × 10 cm × 1.5 cm square Petri dish (LDP [Wayne, New Jersey, USA] D210-16). Plates were incubated for 3 days at 25˚C in the dark and imaged using a flatbed scanner or VHX-1000 digital microscope (Keyence, Japan).

## Colony-forming unit assay

Macrocolony biofilms were grown for 3 days on 1% tryptone and 1% agar. Each biofilm was transferred to a 2-ml tube with a sealing O-ring (Benchmark Scientific D1031-T20) containing 1 ml of phosphate-buffered saline and 0.5 g of ceramic beads (Fisher Scientific 15-340-159). Samples were homogenized using an Omni Bead Ruptor 12 (Omni 19–050) on the "high" setting for 90 s at 4˚C. Samples were left on ice for 10 min to allow bubbles to clear, and then diluted in PBS by a factor of $10^{-6}$. Ten microliters were spotted on 1% tryptone, 1.5% agar plates, which were tilted vertically to allow the spot to run down the plate and thereby increase the growth area. After 16 h of growth at 37˚C, single colonies were counted and recorded.

## Paraffin embedding, thin sectioning, and confocal imaging

After 3 days of growth as described above, biofilms were overlaid with 1% agar and sandwiched biofilms were lifted from the bottom layer and fixed overnight in 4% paraformaldehyde in PBS at 25˚C for 24 h in the dark. Fixed biofilms were washed twice in PBS and dehydrated through a series of 60-min ethanol washes (25%, 50%, 70%, 95%, 3 × 100% ethanol) and cleared via three 60-min incubations in Histoclear-II (National Diagnostics); these steps were performed using an STP120 Tissue Processor (Thermo Fisher Scientific). Biofilms were then infiltrated with wax via 2 separate 2-h washes of 100% paraffin wax (Paraplast Xtra) at 55˚C and allowed to polymerize overnight at 4˚C. Trimmed blocks were sectioned in 10-μm thick sections perpendicular to the plane of the biofilm, floated onto water at 42˚C, and collected onto slides. Slides were air-dried overnight, heat-fixed on a hotplate for 30 min at 45˚C, and rehydrated in the reverse order of processing. Rehydrated colonies were immediately mounted in (Thermo Fisher Scientific ProLong Diamond Antifade Mountant P36965) and overlaid with a coverslip. Fluorescent images were captured using an LSM800 confocal microscope (Zeiss, Germany). Samples were illuminated with 488 and 561 nm lasers, respectively. eGFP acquisition parameters were: 400 to 548 nm (emission wavelength range). mScarlet acquisition parameters were: 593 to 700 nm (emission wavelength range). Each image constitutes a composite ("Z-stack") of images taken at 5 focal planes at 1 μm intervals. Each strain was prepared in this manner in at least biological triplicates.

## Image analysis and quantification of biofilm microorganization

Image analysis was performed in Fiji [116] and Python (Python Software Foundation; Python Language Reference, version 3.8. Available at www.python.org). Each image was viewed in Fiji for quality control and then binarized using a custom Python script. For each sample, a fixed-threshold binarization was applied to the fluorescence image using a multilevel version of Otsu's method, which separates pixels of an input image into several classes obtained according to intensity [117]. For most images, the number of classes $n$ was set to 3, and the image was

binarized by setting pixels in the brightest class equal to 1 and all others to 0. Qualitative inspection of the thresholded images was performed for quality assurance; samples which yielded poor separation into 3 classes (due to, e.g., lower resolution or fluorescence intensity) were binarized using the classic Otsu's method [118]. Images were then cleaned by using masks to remove artifacts. All successive analysis and quantification was performed using the cleaned binarized images.

Boundaries of individual cells were segmented from binarized images using the Python SciPy package [119]. The orientation of each cell was determined as the angle between the horizontal and the major axis of the best-fit ellipse to each cell, ranging from −90 to +90 degrees. Heights along the biofilm's direction of growth were divided into bins of 25 pixels in width. An order parameter to quantify cell alignment was defined as the standard deviation of the orientation of all cells within a bin; a high value indicates that cells in a bin are randomly oriented, while a low value indicates that cells are aligned in the same direction. All code is available at https://github.com/jnirody/biofilms.

### Pellicle biofilm mixing assay

Overnight precultures of fluorescent (mScarlet-expressing) and non-fluorescent strains were diluted 1:100 in LB and grown to mid-exponential phase (OD at 500 nm ~ 0.5). OD values at 500 nm were read in a Spectronic 20D+ spectrophotometer (Thermo Fisher Scientific) and cultures were adjusted to the same OD. Adjusted cultures were then mixed in a 2.5:97.5 ratio of fluorescent:non-fluorescent cells and 9 ml of this mixture was added to $13 \times 100$ mm culture tubes (VWR, Cat no. 10545–936) and grown in 25°C incubator for 3 days. Pellicles were removed from glass tubes using a sterile needle, transferred to plates and overlayed with 1% agar. Samples were subsequently processed using our standard paraffin embedding and thin sectioning protocol.

### Fixed macrocolony stimulated Raman scattering microscopy

Five microliters of a bacterial subculture were spotted onto 45 ml/15 ml two-layer 1% tryptone 1% agar plates and were grown for 72 h, then the top layer and biofilm was transferred to 2 ml 1% tryptone 1% agar 50% $D_2O$ (in a $35 \times 10$ mm Petri dish) and incubated at 25°C for 5 h or 24 h. The biofilms were then thin sectioned as described above. For SRS microscopy, an integrated laser source (picoEMERALD, Applied Physics & Electronics) was used to produce both a Stokes beam (1,064 nm, 6 ps, intensity modulated at 8 MHz) and a tunable pump beam (720 to 990 nm, 5 to 6 ps) at an 80 MHz repetition rate. The spectral resolution of SRS is FWHM = 6–7 $cm^{-1}$. These 2 spatially and temporally overlapped beams with optimized near-IR throughput were coupled into an inverted multiphoton laser-scanning microscope (FV1200MPE, Olympus). Both beams were focused on the biofilm samples through a 25× water objective (XLPlan N, 1.05 N.A. MP, Olympus) and collected with a high N.A. oil condenser lens (1.4 N.A., Olympus) after the sample. By removing the Stokes beam with a high O.D. bandpass filter (890/220 CARS, Chroma Technology), the pump beam is detected with a large area Si photodiode (FDS1010, Thorlabs) reverse-biased by 64 DC voltage. The output current of the photodiode was electronically filtered (KR 2724, KR electronics), terminated with 50Ω, and demodulated with a lock-in amplifier (HF2LI, Zurich Instruments) to achieve near shot-noise-limited sensitivity. The stimulated Raman loss signal at each pixel was sent to the analog interface box (FV10- ANALOG, Olympus) of the microscope to generate the image. All images were acquired with 80 μs time constant at the lock-in amplifier and 100 μs pixel dwell time (approximately 27 s per frame of $512 \times 512$ pixels). Measured after the objectives, 40 mW pump power and 120 mW Stokes beam were used to image the carbon-deuterium 2,183 $cm^{-1}$ and off-resonance 2,004 $cm^{-1}$ channels.

### Live macrocolony D$_2$O quantification

Five microliters of a bacterial subculture were spotted onto a bilayer plate of 50 mL (bottom layer) and 10 mL (top layer) of 1% tryptone 1% agar [Teknova (Hollister, CA) A7777] in a 10 cm x 10 cm × 1.5 cm square Petri dish (LDP [Wayne, NJ] D210-16). Plates were grown for 48 h, at 25˚C in the dark. The top layer and biofilm were then transferred onto another bilayer plate of 50 mL (bottom layer) and 10 mL (top layer) of 1% tryptone, 1% agar [Teknova (Hollister, CA) A7777] containing 50% D$_2$O in a 10 cm × 10 cm × 1.5 cm square Petri dish (LDP [Wayne, NJ] D210-16) and incubated at 25˚C for 1 h. The biofilms were then transferred to slides with 0.6-mm thick spacers (Sigma-Aldrich BBL665504-25EA) before placing a coverslip on top for imaging. For D$_2$O concentration, 40 mW pump power and 120 mW Stokes power (measured after the objectives) was used to image the oxygen-deuterium 2,489 cm$^{-1}$ and the protein CH3 2,940 cm$^{-1}$ channels. These images were acquired with 4 μs time constant at the lock-in amplifier and 4 μs pixel dwell time (approximately 1.6 s per frame of 512 × 512 pixels). The SRS image for D$_2$O was generated by normalizing the oxygen-deuterium image (2,489 cm$^{-1}$) against the protein CH3 image (2,940 cm$^{-1}$) to correct for light scattering.

### Colony morphology assay

Precultures grown for 14 to 16 h were subcultured 1:100 and grown until OD at 500 nm of approximately 0.5, and 10 μl were spotted onto 60 ml of colony morphology medium (1% tryptone, 1% agar [Teknova, A7777] containing 40 μg/ml Congo red dye [VWR, AAAB24310-14] and 20 μg/ml Coomassie blue dye [VWR, EM-3300]) in a 10 cm × 10 cm × 1.5 cm plates. Plates were incubated for up to 5 days at 25˚C with >90% humidity and imaged using a flatbed scanner [Epson, E11000XL-GA].

### Macrocolony clearing and light sheet microscopy

Overnight precultures were diluted 1:100 in LB and grown to mid-exponential phase (OD at 500 nm ~ 0.5). OD values at 500 nm were read in a Spectronic 20D+ spectrophotometer (Thermo Fisher Scientific) and cultures were adjusted to the same OD. Adjusted cultures were then mixed in a 1:100 ratio of fluorescent:non-fluorescent cells; 0.5 microliters of mixture were spotted onto a bilayer plate of 50 ml (bottom layer) and 10 ml (top layer) of 1% tryptone, 1% agar [Teknova (Hollister, California, USA) A7777] in a 10 cm × 10 cm × 1.5 cm square Petri dish (LDP [Wayne, New Jersey, USA] D210-16). Plates were incubated for 1 day at 25˚C in the dark. Biofilms were overlaid with 1% agar, and sandwiched biofilms were lifted from the bottom layer, and fixed overnight in 4% paraformaldehyde in PBS at 25˚C for 12 h in the dark. Fixed biofilms were washed twice in PBS then incubated for 3 h in 70% (V/V) glycerol solution (with 2.5 mg/ml 1,4-diazabicyclo[2.2.2]octane (DABCO)) deionized water solution for optical clearing. The clarified biofilms were mounted on a standard microscopy slide and were imaged with ClearScope [120] (MBF Biosciences) light sheet microscope, using Nikon 20×/1.0NA detection objective. The resulting image tiles were stitched by using a Terastitcher-based custom pipeline [121,122] and analyzed using ImageJ after 2-fold down-sampling in x-y axes.

### Sheared-cell SDS-PAGE and western blotting

Sheared pilin and cell lysate proteins were extracted as previously described [123] with some modifications. Overnight precultures were diluted 1:100 in LB and grown to mid-exponential phase (OD at 500 nm ~ 0.5). OD values at 500 nm values were read in a Spectronic 20D+ spectrophotometer (Thermo Fisher Scientific) and cultures were adjusted to the same OD. Five microliters of culture were spotted onto a plate of 60 ml of 1% tryptone, 1% agar [Teknova

(Hollister, California, USA) A7777] in a 10 cm × 10 cm × 1.5 cm square Petri dish (LDP [Wayne, New Jersey, USA] D210-16). Plates were incubated for 3 days at 25°C in the dark.

Biofilm samples (25 biofilms per strain) were gently scraped from agar, resuspended in 1 ml sterile PBS (phosphate buffered saline, pH 7.4), and centrifuged at $4,000 \times g$ for 5 min at room temperature. Pellets were then resuspended in 4.5 ml of sterile PBS and vortexed for 30 s to shear cell surface proteins and pilin. After vortexing, sheared cells were centrifuged at $4,000 \times g$ for 15 min at room temperature and the supernatant containing the sheared surface pilin was discarded.

Cellular proteins were extracted from the cell pellets after surface pilin was sheared and removed (described above). Briefly, cell pellets were resuspended in 0.2 M Tris-Cl (pH 8.0), normalized to OD at 600 nm ~ 4.0, centrifuged, and pellets were resuspended in 1× LDS sample buffer with 5% β-mercaptoethanol. Samples were boiled for 5 min, centrifuged, and aliquots from the supernatants were resolved on a 4% to 12% NuPAGE Bis-Tris gel for visualization and immunoblotting.

For PilA immunoblotting, both sheared and whole-cell pilin preparations were resolved using 4% to 12% NuPAGE Bis-Tris gels at 125 V for 2.5 h and transferred onto 0.22 μm nitrocellulose membranes. Protein bands were normalized to total protein visualized by BLOT-FastStain (GBiosciences). Membranes were blocked using TBS-T (Tris-buffered saline with Tween-20) containing 5% milk before probing with an anti-PilA rabbit polyclonal antibody (gift of the O'Toole lab, Dartmouth) [124] at 1:1,000, followed by an IR-conjugated goat anti-rabbit antibody at 1:10,000 (LI-COR Biosciences). Bands were visualized using the LI-COR Odyssey Clx imager and Image Studio. Uncropped blots are shown in **S6A Fig**.

## Whole cell SDS-PAGE and western blotting

Overnight cultures were diluted 1:100 in LB and grown to mid-exponential phase (OD at 500 nm ~0.5). OD measurements were made in a Spectronic 20D+ spectrophotometer (Thermo Fisher Scientific) and cultures were adjusted to the same OD. Five microliters of culture were spotted onto a plate of 60 ml of 1% tryptone, 1% agar [Teknova (Hollister, California, USA) A7777] in a 10 cm × 10 cm ×1.5 cm square Petri dish (LDP [Wayne, New Jersey, USA] D210-16). Plates were incubated for 3 days at 25°C in the dark.

Biofilm samples (25 biofilms per strain) were gently scraped from agar, resuspended in 1 ml sterile PBS (pH 7.4) and centrifuged at $4,000 \times g$ for 10 min at 4°C. Cell pellets were resuspended in 0.2 M Tris-Cl (pH 8.0), normalized to OD at 600 nm ~ 4.0, centrifuged at $16,000 \times g$ at 4°C for 5 min. These washes were repeated a total of 3 times. Pellets were resuspended in 1× LDS sample buffer with 5% β-mercaptoethanol. Samples were boiled for 5 min, centrifuged, and aliquots from the supernatants were resolved on a 4% to 12% NuPAGE Bis-Tris gel for visualization and immunoblotting.

PilA immunoblotting was performed as described above.

## Biofilm rhamnose-induction assay

Overnight precultures were diluted 1:100 in LB and grown to mid-exponential phase (OD at 500 nm ~ 0.5). OD values at 500 nm were read in a Spectronic 20D+ spectrophotometer (Thermo Fisher Scientific) and cultures were adjusted to the same OD. Adjusted cultures were then mixed in a 2.5:97.5 ratio of constitutively fluorescent eGFP cells:rhamnose inducible mScarlet fluorescent cells. Five microliters of mixture were spotted onto bilayer plates. Plates were incubated for 3 days at 25°C in the dark, and then transferred to medium containing 0.5% L-rhamnose for 5 h. Samples were subsequently processed using our standard paraffin embedding and thin sectioning protocol.

## Liquid-culture growth and rhamnose-induction assay

Overnight precultures in biological triplicate were diluted 1:100 into 1% tryptone with 0.5% L-rhamnose when indicated in 96-well, black-sided, clear-bottomed plate [Greiner Bio-One; Millipore Sigma, M5811] and incubated at 37˚C with continuous shaking on the medium setting in a Biotek Synergy H1 plate reader. Growth was assessed by taking OD readings at 500 nm and mScarlet readings at excitation and emission wavelengths of 569 nm and 599 nm, respectively, every 30 min for 24 h.

## Live-imaging of dye distribution in pellicle biofilms

Liquid cultures were grown in LB at 37˚C with shaking at 250 rpm. Overnight precultures were diluted 1:100 in LB and grown to mid-exponential phase (OD at 500 nm of ~ 0.5). OD values at 500 nm were read in a Spectronic 20D+ spectrophotometer (Thermo Fisher Scientific) and cultures were adjusted to the same OD. Adjusted cultures were then mixed in a 2.5 to 97.5 ratio of fluorescent to non-fluorescent cells. Two ml of mixed culture was added to a 10-mm optical glass Type 1FL Macro Fluorescence Cuvette (1FLG10, Fireflysci). The cuvette was covered with parafilm and incubated at 25˚C for 3 days. Five μl of 20 μm Sulfo-Cy5 NHS-ester (Lumiprobe #23320 (diluted in water)) was pipetted through the pellicle biofilm into the liquid medium. Live imaging was performed with projected Light Sheet Microscopy (pLSM) [78] using ASI 16.67X/0.4 NA objective and 4 pixels-thick light sheet. Samples were excited with lasers at 488 nm for eGFP and 647 nm for the Sulfo-Cy5 NHS-ester dye lasers and imaged using a Quad Band Emission Filter (432/515/595/730 nm, Semrock).

## Microsphere assay

Overnight precultures were diluted 1:100 in LB and grown to mid-exponential phase (OD at 500 nm of ~ 0.5). OD values at 500 nm were read in a Spectronic 20D+ spectrophotometer (Thermo Fisher Scientific) and cultures were adjusted to the same OD. Adjusted cultures were then mixed in a 2.5:97.5 ratio of fluorescent:non-fluorescent cells. Bilayer plates of 45 ml (bottom layer) and 15 ml (top layer) of 1% tryptone, 1% agar [Teknova (Hollister, California, USA) A7777] in a 10 cm × 10 cm × 1.5 cm square Petri dish (LDP [Wayne, New Jersey, USA] D210-16) were made and solidified overnight; 300 μl (10^10 microspheres/ml) of Fluoresbrite Multifluorescent Microspheres 0.2 μm (PolySciences 24050–5) were spread onto a bilayer plate using sterile glass beads and dried for 30 min. Five microliters of bacterial culture mixture were spotted onto plate and incubated for 3 days at 25˚C in the dark and macrocolony biofilms were subsequently imaged live or prepared for cryosectioning. Samples were imaged on a ZEISS Axio Zoom.V16 microscope, equipped with a mercury lamp. eGFP was detected using a 450/490 nm excitation filter, 500/550 nm emission filter, and 495 nm dichroic mirror. Fluoresbrite Multifluorescent Microspheres were detected using a 538/562 nm excitation filter, 570/640 nm emission filter, and 570 nm dichroic mirror.

## Macrocolony biofilm cryosectioning

Macrocolony biofilms were overlaid with 1% agar, and sandwiched biofilms were cut and lifted from the bottom layer, and placed in disposable cryomold (Tissue-Tek). They were then covered with an embedding agent (Tissue-Tek OCT) and flash frozen in liquid nitrogen. Cryoembedded biofilms were subsequently sectioned with a Leica CM1950 cryostat set at −18˚C. Ten micrometer-thick sections perpendicular to the plane of the macrocolony were then transferred onto slides and immediately imaged using a ZEISS Axio Zoom.V16.

### Propidium iodide and antibiotic assay

Overnight precultures were diluted 1:100 in LB and grown to mid-exponential phase (OD at 500 nm of ~ 0.5). OD values at 500 nm were read in a Spectronic 20D+ spectrophotometer (Thermo Fisher Scientific) and cultures were adjusted to the same OD. Adjusted cultures were then mixed in a 2.5-to-97.5 ratio of eGFP fluorescent:non-fluorescent cells. Five microliters of mixture were spotted onto bilayer plate. Plates were incubated for 3 days at 25˚C in the dark, and then transferred to medium containing 50 μm PI with and without 100 μg/ml tobramycin for 5 h. Samples were subsequently processed using standard paraffin embedding and thin sectioning protocol. Samples were imaged on a ZEISS Axio Zoom.V16 microscope, equipped with a mercury lamp. PI staining was detected using a 538/562 nm excitation filter, 570/640 nm emission filter, and 570 nm dichroic mirror.

## Supporting information

**S1 Fig. Expression of mScarlet does not impact fitness in macrocolony biofilms.** PA14 WT constitutively expressing mScarlet (2.5%) was mixed with PA14 WT (97.5%), spotted on 1% tryptone 1% agar plates, and grown for 3 days. The percentages of fluorescent cells were determined after 3 days of growth by homogenizing the macrocolonies and plating for colony-forming units (CFUs). Approximately 2.5% of CFUs expressed mScarlet. Results for 4 biological replicates are shown. The data underlying this figure can be found in S1_raw_data.
(PDF)

**S2 Fig. D$_2$O distribution is uniform in macrocolony biofilms.** Signal across depth detected for D$_2$O in live two-day-old PA14 biofilms imaged by SRS. D$_2$O is normalized to total protein. Results are shown for 3 biological replicates. The data underlying this figure can be found in S1_raw_data.
(PDF)

**S3 Fig. Anoxic conditions abrogate striation formation.** Fluorescence micrographs of thin sections from WT biofilms grown on 1% tryptone, 1% agar under oxic or anoxic conditions for 3 days. Where indicated, 40 mM potassium nitrate was included in the growth medium. Biofilm inocula contained 2.5% cells that constitutively express mScarlet. mScarlet fluorescence is colored yellow. Scale bar is 25 μm and applies to all images.
(PDF)

**S4 Fig. Effects on cellular arrangement do not correlate with consistent effects on macrocolony morphology between mutants.** Biofilms were grown for 72 h on 1% tryptone, 1% agar medium containing the dyes Congo red and Coomassie blue. Scale bar applies to all images.
(PDF)

**S5 Fig. Mutations affecting global regulators, pilus synthesis, and O-antigen synthesis/attachment alter cell patterning in colony biofilms.** Fluorescence micrographs of thin sections from indicated mutant biofilms grown on 1% tryptone and 1% agar for 3 days. The biofilm inocula contained constitutive mScarlet-expressers at a frequency of 2.5%; 97.5% of the cells did not express a fluorophore. Scale bar is 25 μm. Images are representative of at least 2 independent experiments and mScarlet fluorescence is colored yellow. Scale bar applies to all images.
(PDF)

**S6 Fig. Cell-arrangement phenotypes are not due to changes in PilA levels.** (**A**) Uncropped version of western blot shown in **Fig 5B** indicating PilA protein levels from sheared cells for macrocolony biofilms of WT and the indicated mutants. (**B**) Cropped western blot showing

PilA protein levels from whole cells for macrocolony biofilms of WT and the indicated mutants. (**C**) Uncropped version of western blot shown in (B). (**A–C**) For each gel, equal amounts of total protein were loaded per lane and resolved by SDS-PAGE using a 15% poly-acrylamide gel. The PilA protein was detected using an anti-PilA antibody.
(PDF)

**S7 Fig. Δ*gacA* biofilms show increased rhamnose-dependent expression at the biofilm–air interface, consistent with their disordered cellular arrangement.** Fluorescence micrographs of thin sections from WT and indicated mutant biofilms. Biofilm inocula contained 2.5% cells that constitutively express eGFP and 97.5% RhaSR-P*rhaBAD*-controlled mScarlet-producing strain. RhaSR-P*rhaBAD*-controlled mScarlet was induced with 0.5% L-rhamnose. Top panels show eGFP fluorescence (colored yellow) and bottom panels show the mScarlet fluorescence for each thin section. Quantification of mScarlet fluorescence is shown on the right. Scale bar is 25 μm and applies to all images. The data underlying this figure can be found in S1_raw_-data.
(PDF)

**S8 Fig. RhaSR-P*rhaBAD*-controlled mScarlet production in WT, Δ*wbpM*, Δ*pilA*, Δ*wbpM* Δ*pilA* strains is comparable during planktonic growth.** Changes in fluorescence (top) and optical density (bottom) for the indicated reporter strains during growth in liquid culture with or without the addition of 0.5% L-rhamnose. The data underlying this figure can be found in S1_raw_data.
(PDF)

**S9 Fig. Dye distribution in live pellicle biofilm.** (**A**) Side view of a WT *P. aeruginosa* pellicle biofilm grown for 3 days in a cuvette. WT and Δ*wbpM* form comparable pellicles. (**B**) Live light sheet microscopy images of WT and Δ*wbpM* mixing assay pellicle biofilms. Scale bar applies to both images. (**C**) Quantification of Sulfo-Cy5 NHS-ester fluorescence after addition to the medium below each pellicle biofilm. The signal was normalized by total sum. The data underlying this figure can be found in S1_raw_data.
(PDF)

**S10 Fig. Metabolic activity profile within Δ*ssg* biofilm.** SRS micrographs of WT and Δ*ssg* biofilm thin sections. Scale bar is 25 μm and applies to both images. The data underlying this figure can be found in S1_raw_data.
(PDF)

**S11 Fig. Mutations affecting cellular arrangement alter biofilm antibiotic tolerance profiles.** (**A**) Fluorescence micrographs of thin sections from indicated mutant biofilms grown on 1% tryptone and 1% agar for 3 days. The biofilm inocula contained 2.5% of cells that constitutively express mScarlet (shown as yellow); 97.5% did not express a fluorophore. Scale bar is 25 μm and applies to all images. (**B**) Schematic illustration of the experimental setup for growing *P. aeruginosa* biofilms on agar plates and their subsequent transfer to a medium containing 50 μm propidium iodide or 50 μm propidium iodide + 100 μg/ml tobramycin. (**C**) **Left**: Fluorescence micrographs of 3-day-old biofilms exposed to propidium iodide. **Center**: Fluorescence micrographs of biofilms treated with tobramycin and PI. **Right**: Graphs showing quantification of PI staining. Images shown in this figure are representative of at least 2 independent experiments. The data underlying this figure can be found in S1_raw_data.
(PDF)

**S1 Table. Bacterial strains used in this study.**
(PDF)

**S2 Table. Plasmids used in this study.**
(PDF)

**S3 Table. Primers used in this study.**
(PDF)

**S1 Raw images. Cell-arrangement phenotypes are not due to changes in PilA levels.** (**A**) Uncropped version of western blot shown in **Fig 5B** indicating PilA protein levels from sheared cells for macrocolony biofilms of WT and the indicated mutants. (**B**) Cropped western blot showing PilA protein levels from whole cells for macrocolony biofilms of WT and the indicated mutants. (**C**) Uncropped version of western blot shown in **S6B Fig**. For each gel, equal amounts of total protein were loaded per lane and resolved by SDS-PAGE using a 15% polyacrylamide gel. The PilA protein was detected using an anti-PilA antibody.
(PDF)

**S1 Data. Spreadsheet with raw data for graphs in Figs 2B, 6D, 6F, 7B, 7D, S2, S7, S8, S9, S10 and S11.**
(XLSX)

## Acknowledgments

The authors thank Endah Rosa, Paula Fernandez Musso, and Yu-Cheng Lin for assistance in generating mutants described in this manuscript; Sherry Kuchma and the O'Toole Lab for gifting the anti-PilA rabbit polyclonal antibody; and Riley Gentry and the Gonzalez Lab for gifting the dye Sulfo-Cy5 NHS-ester.

## Author Contributions

**Conceptualization:** Hannah Dayton, William Cole Cornell, Chase J. Morgan, Alexa Price-Whelan, Lars E. P. Dietrich.

**Data curation:** Hannah Dayton, Shradha Chauhan, Emily LaMarre, William Cole Cornell, Jasmine A. Nirody, Lars E. P. Dietrich.

**Formal analysis:** Hannah Dayton, Mian Wei, Shradha Chauhan, Jasmine A. Nirody, Lars E. P. Dietrich.

**Funding acquisition:** Anuradha Janakiraman, Wei Min, Raju Tomer, Jasmine A. Nirody, Lars E. P. Dietrich.

**Investigation:** Hannah Dayton, Julie Kiss, Mian Wei, Emily LaMarre, Chase J. Morgan, Lars E. P. Dietrich.

**Methodology:** Hannah Dayton, Julie Kiss, Mian Wei, Shradha Chauhan, Emily LaMarre, William Cole Cornell, Chase J. Morgan, Raju Tomer.

**Project administration:** Hannah Dayton, Lars E. P. Dietrich.

**Resources:** Anuradha Janakiraman, Wei Min, Raju Tomer, Lars E. P. Dietrich.

**Software:** Jasmine A. Nirody.

**Supervision:** Alexa Price-Whelan, Lars E. P. Dietrich.

**Validation:** Hannah Dayton, Lars E. P. Dietrich.

**Visualization:** Hannah Dayton, Alexa Price-Whelan, Lars E. P. Dietrich.

**Writing – original draft:** Hannah Dayton, Alexa Price-Whelan, Lars E. P. Dietrich.

**Writing – review & editing:** Hannah Dayton, Julie Kiss, Mian Wei, Shradha Chauhan, Emily LaMarre, William Cole Cornell, Chase J. Morgan, Anuradha Janakiraman, Wei Min, Raju Tomer, Alexa Price-Whelan, Jasmine A. Nirody, Lars E. P. Dietrich.

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
