## [Editor Report · Decision Letter 0]

15 Jun 2023

Dear Dr. Dietrich, 

Thank you for submitting your manuscript entitled "Cell arrangement impacts metabolic activity and antibiotic tolerance in Pseudomonas aeruginosa biofilms" for consideration as a Research Article by PLOS Biology.

Your manuscript has now been evaluated by the PLOS Biology editorial staff, as well as by an academic editor with relevant expertise, and I am writing to let you know that we would like to send your submission out for external peer review.

Once your full submission is complete, your paper will undergo a series of checks in preparation for peer review. After your manuscript has passed the checks it will be sent out for review. To provide the metadata for your submission, please Login to Editorial Manager (https://www.editorialmanager.com/pbiology) within two working days, i.e. by Jun 17 2023 11:59PM.

Kind regards,

Paula

---

Senior Editor

PLOS Biology

---

## [Decision Letter · Decision Letter 1]

29 Aug 2023

Dear Dr. Dietrich,

Thank you for your patience while your manuscript "Cell arrangement impacts metabolic activity and antibiotic tolerance in Pseudomonas aeruginosa biofilms" was peer-reviewed at PLOS Biology. It has now been evaluated by the PLOS Biology editors, an Academic Editor with relevant expertise, and by several independent reviewers. 

In light of the reviews, which you will find at the end of this email, we would like to invite you to revise the work to thoroughly address the reviewers' reports.

As you will see below, the reviewers agree that the work is interesting, however they all find issues that will need to be addressed before further consideration at PLOS Biology. In particular, we think that it is important to improve the mechanistic interpretation of your observations and to better place them into the context of existing literature. Please address all the reviewers' issues. 

Given the extent of revision needed, we cannot make a decision about publication until we have seen the revised manuscript and your response to the reviewers' comments. Your revised manuscript is likely to be sent for further evaluation by all or a subset of the reviewers.

**IMPORTANT - SUBMITTING YOUR REVISION**

*Re-submission Checklist*

*Published Peer Review*

*PLOS Data Policy*

*Blot and Gel Data Policy*

Sincerely,

Paula

---

Senior Editor

PLOS Biology

REVIEWS:

Reviewer #1: Microbial biofilms.

Reviewer #2: Akos T Kovacs. Bacterial interaction and evolution.

Reviewer #3: P. aeruginosa communities and biofilms.

Reviewer #1: 

In the manuscript "Cell arrangement impacts metabolic activity and antibiotic tolerance in Pseudomonas aeruginosa biofilms," Dayton et al. conducted a detailed investigation into the cellular organization within P. aeruginosa colony biofilms. The authors explored the role of various cell factors, including adhesins, sigma factors, second messenger signalling factors, and metabolic factors, in determining a distinctive pattern of cellular arrangement. They also examined how this pattern influenced the metabolic activity and survival capacity of cells when exposed to antibiotics. Specifically, the authors found that in an internal zone (zone 2) cells form long vertically oriented chains referred to as "striations" and identified cell surface components (pilus and O-antigen) that impacted such pattern of cellular organization across the biofilm depth. Using stimulated Raman scattering microscopy and dead cell labelling, the authors described relationships between cellular arrangement, cell metabolic status, and susceptibility to antibiotic. Overall, it is very nice study that provides novel insights into an aspect of bacterial biofilms that has not been extensively explored, which is the role of cellular organization in biofilm physiology. I think this work can be of great interest for researchers in cell biology and, particularly, for the biofilm researcher community. The study is well-conducted, with robust data and proper statistical analysis. I have made some comments, which I present below.

Comments:

- The approach used to visualize "striations" in biofilms involved inoculating 2.5% cells that constitutively express mScarlet, representing a 1:40 ratio of mScarlet-labelled cells to non-labelled cells. Upon reviewing the images in the various figures, it appears that the number of mScarlet-expressing cells is higher than the initial 1:40 ratio. Are mScarlet-labelled cells growing faster or out competing non-labelled cells?

- Early in the Result/Discussion section the authors presented data on how the availability of resources (tryptone concentration) affects the organization of cellular-arrangement zones. Later in the same section, they presented data on how cell patterning influences the distribution of substrates (rhamnose and microspheres). Although these subsections appear to be related, the use of different substrates (tryptone vs rhamnose) and terminology (resources vs substrates) creates confusion. The authors should clarify the relationship between these findings and consider revising the text to ensure a more cohesive presentation.

- Related to the data presented in the subsection "Cell patterning influences the distribution of substrates across biofilm depth", I am not sure if the L-rhamnose reporter system and the microspheres are the appropriate systems to evaluate the distribution of substrates across the biofilm. In the case of the rhaSRPrhaBAD inducible promoter system, the authors already indicated that pleiotropic effects inhibited activity of the system in biofilm zones 1 and 2. In the case of the microspheres, given their size I would not consider it as a diffusible substrate. The fact that beads remain at the agar interface in the pilA mutant biofilm (Fig 6H), indeed reflects that beads are somehow limited to enter into the biofilm, but this does not necessarily imply an inability of diffusible substrates to penetrate or distribute differentially. Have the authors considered alternative approaches or additional experiments for a more accurate assessment of substrate distribution within the biofilm?

- Based on the data presented in Figure 7, the authors' interpretation regarding the correlation between cellular arrangement and antibiotic tolerance is understandable. However, it is evident that the zone exhibiting vertical striations is significantly thicker (about 130 μm) than the zone of antibiotic susceptibility (20-60 μm). Although the subzone ranging from 60 to 130 μm displays the same cellular organization pattern as the 20-60 μm subzone, cells are only susceptible within the latter, where high metabolic activity occurs due to optimized oxygen and nutrient availability. It appears that the ordered cells in the 60-130 μm subzone do not prevent the antibiotic from reaching cells above that subzone. Thus, the relationship presented may not be as direct as suggested.

- The manuscript is well-written and structured. However, while the "Results and Discussion" section adequately describes the results and their corresponding conclusions, it would benefit from a more extensive discussion. Although some discussions of the results are provided, the authors should consider expanding on certain aspects not discussed. 

Reviewer #2: This is an exceptional work from the Dietrich lab that demonstrates microanatomy of Pseudomonas aeruginosa colony biofilm and highlights parallel arranged cells that influence metabolism and therefore also specific (grown-dependent) antibiotic sensitivity. The work includes molecular details (mutants) and elegantly performed microscopy approaches. I have only minor comments that should be easily addressed.

Technical point: The authors should explain if mScarlet overexpression has any influence on the fitness of the cells. This might have been described previously or tested by the authors to circumvent any bias caused by the fluorescent marker. When mixing 2.5% of fluorescent marker labeled strains, will the 2.5% roughly remains at the end of the experiments? Such quantification of strain abundance after growth might answer this concern. 

The manuscript is written by a direct style, which I let the editor to decide whether it should be changed or remain as it is, e.g. line 160 "we were surprised", line 320: "we will therefore focus our interpretation"

Line 182: this should be the same for pure cultures also, not only with mixing-assay biofilms, right?

Line 210: it would benefit the readers if the authors would include in the table which mutation influence colony structure or other type of Pseudomonas biofilms, so it provides a good comparison to the microanatomy. The authors conclude that macroscale does not correlate with cellular arrangement, but this is hard to evaluate without knowing the full literature. Adding that background information in the table would make it easier to evaluate. 

Line 229: Is wbpM known to be regulated by GacS/GacA or LasR system? Does this gene display different transcript level in these mutants in previous (biofilm) transcriptomes preformed with gacS/gacA or lasR mutants?

Scale bar should be added to the CLSM images (mixing assays) in Fig 1B and C (I understand that orientation panels practically include the scale, but in later figures, that is not added); Fig 2B, Fig 3B Fig 4B, Fig 5C, Fig 7D

Superb work, congratulations!

ATK

Reviewer #3: In this manuscript, the authors investigate the effect of different vertical cellular arrangements in a colony on agar. The found that depending upon the region of the colony, different arrangements are observed. They identified type IV pili mutants as defective for normal cellular arrangements in colonies. In general, this is an interesting manuscript with some cool observations. But little is provided mechanistically to explain many of the observations that have been made, reducing my enthusiasm for the study.

* The introduction is unusually vague and brief. I would restructure it to inform the readers about what is known regarding biofilm structure and the environmental factors that influence it, paying attention to the many excellent papers measuring vertical nutrient gradients in P. aeruginosa colony biofilms (PS Stewart and others).

* Related to the data shown in figure 2. The inverse gradients of O2 (High to low from colony top to bottom) and nutrients from the agar (high to low from colony bottom to top) make the data presented here hard to interpret. I think the authors should grow their colony biofilms anaerobically on agar containing nitrate, and see if the trends they have observed hold up. By using nitrate anaerobically, the key gradients would be aligned vertically from colony bottom to top (for terminal electron acceptor and nutrients).

* I think it is very important for the authors to demonstrate that there is not a gradient of D2O availability vertically across the colonies. D2O bioavailability in different regions of the colony could skew the signals observed in Fig 2B.

* Sentence lines 216-219 is unclear and should be reworded.

* What are the metabolic activity gradients for the different mutant arrangements shown in Fig 3? 

* Line 248. Fig 3D??

* Lines 276-278 Reports from the Diggle/Goldberg labs have already shown that LPS alterations can influence cellular distribution and packing.

* The major approach that supports most of the authors conclusions are based on microscopy applied to "mixing" experiments. It would be much more convincing if there were additional lines of experimentation or methodologies to support and complement the mixing experiments.

* For the rhamnose-inducible mScarlet, the authors should show in liquid culture that the mutants have the same rhamnose-induced induction curves- a minor point, but a good control experiment.

* The observation that PilA is required for cellular arrangements of the O-antigen mutants is interesting (Fig 5). However, there is no convincing mechanistic explanation for this, making these data very phenomenological.

* Fig 6B- its not clear to me why there are vertical striations in some of the MScarlet expression profiles !?

* The data presented in Fig. 6 F-H are pretty cool, with the pilA mutations affecting microsphere distribution, but the mechanism responsible for this is unclear and the authors suggestion as to why vague. Im not sure at all what this result means.

* The data shown in Fig 7D have been demonstrated previously by a number of groups- zones of metabolic activity correspond to zones on antimicrobial sensitivity. 

* Why is a wbpM mutant showing uniform PI- staining in the colony!?

---

## [Decision Letter · Decision Letter 2]

8 Dec 2023

Dear Dr Dietrich,

My name is Luke Smith - I am an editor at PLOS Biology and have recently taken over handling your manuscript "Cell arrangement impacts metabolic activity and antibiotic tolerance in Pseudomonas aeruginosa biofilms" from my colleague Paula Juaregui, who recently left PLOS for a new job. Thank you for your patience while we considered your revised manuscript. Your revision has now been evaluated by the PLOS Biology editors, the Academic Editor and the original reviewers, who are fully satisfied by the revision (comments below).

Based on the reviews, we are likely to accept your manuscript for publication - however before we can editorially accept your study we need you to address a number of editorial requests, detailed below. 

**IMPORTANT: Please address the following editorial requests: 

1) TITLE: After some discussion within the team, we think the title might be clearer if changed to "Cellular arrangement impacts metabolic activity and antibiotic tolerance in Pseudomonas aeruginosa biofilms". If you agree, please change it accordingly. 

2) Thank you for providing the underlying data related to your figures as a supplemental excel file. A couple of requests: 

a. please reference this file in every relevant figure legend (for example, you can add the sentence "the data underlying this figure can be found in S1_raw_data")

b. I noticed that this file seems to be missing some relevant data. Can you update it to include the raw data related to Figure S1? Also, please add data from orientation plots to this file (ex related to data presented in figures 1B-C; 2B; Fig 3B; Fig 4B; Fig 5B) 

3) METHODS: I noticed that some of the methods are presented as a supplementary file. Can you please move those details into the main text of the manuscript along with the rest of the methods? 

We expect to receive your revised manuscript within two weeks - although happy to provide an extension as needed, given the upcoming holidays. 

*Published Peer Review History*

*Press*

Sincerely,

Luke

Lucas Smith, Ph.D.

Senior Editor,

lsmith@plos.org,

PLOS Biology

Reviewer remarks:

Reviewer #1: I have reviewed the resubmitted revised version of the manuscript "Cell arrangement impacts metabolic activity and antibiotic tolerance in Pseudomonas aeruginosa biofilms" by Dayton et al. and have considered their responses to my comments. 

The authors have addressed all of my comments, incorporating additional data (e.g., Fig. S1 and Fig. S9) to strengthen the findings presented in the previous version of the manuscript and to support their responses. I believe that the overall changes have improved the paper.

I have no further comments.

Reviewer #2, Akos T Kovacs (note, Reviewer 2 has signed this review): Thank you for addressing all comment thoroughly. Congratulations on the great work!

Reviewer #3: Very impressed at the thorough consideration of previous comments given by the author. Although there is mechanistic understanding missing in spots, this manuscript is a very thought provoking, important study.

---

## [Editor Report · Decision Letter 3]

19 Dec 2023

Dear Dr Dietrich,

Thank you for the submission of your revised Research Article "Cellular arrangement impacts metabolic activity and antibiotic tolerance in Pseudomonas aeruginosa biofilms" for publication in PLOS Biology. On behalf of my colleagues and the Academic Editor, Victor Sourjik, I am pleased to say that we can in principle accept your manuscript for publication, provided you address any remaining formatting and reporting issues. These will be detailed in an email you should receive within 2-3 business days from our colleagues in the journal operations team; no action is required from you until then. Please note that we will not be able to formally accept your manuscript and schedule it for publication until you have completed any requested changes.

PRESS

Sincerely, 

Lucas Smith, Ph.D.

Senior Editor

PLOS Biology

lsmith@plos.org